# Learning from Label Proportions: Bootstrapping Supervised Learners via Belief Propagation

**Shreyas Havaldar**[1*]   **Navodita Sharma**[1*]
**Shubhi Sareen**[2]   **Karthikeyan Shanmugam**[1]   **Aravindan Raghuveer**[1]
[1]Google Research India   [2]Google India

## Abstract

Learning from Label Proportions (LLP) is a learning problem where only aggregate level labels are available for groups of instances, called bags, during training, and the aim is to get the best performance at the instance-level on the test data. This setting arises in domains like advertising and medicine due to privacy considerations. We propose a novel algorithmic framework for this problem that iteratively performs two main steps. For the first step (Pseudo Labeling) in every iteration, we define a Gibbs distribution over binary instance labels that incorporates a) covariate information through the constraint that instances with similar covariates should have similar labels and b) the bag level aggregated label. We then use Belief Propagation (BP) to marginalize the Gibbs distribution to obtain pseudo labels. In the second step (Embedding Refinement), we use the pseudo labels to provide supervision for a learner that yields a better embedding. Further, we iterate on the two steps again by using the second step's embeddings as new covariates for the next iteration. In the final iteration, a classifier is trained using the pseudo labels. Our algorithm displays strong gains against several SOTA baselines (upto **15%**) for the LLP Binary Classification problem on various dataset types - tabular and Image. We achieve these improvements with minimal computational overhead above standard supervised learning due to Belief Propagation, for large bag sizes, even for a million samples.

## 1 Introduction

Learning from Label Proportions (henceforth LLP) has seen renewed interest in recent times due to the rising concerns of privacy and leakage of sensitive information (Ardehaly & Culotta, 2017; Busa-Fekete et al., 2023; Zhang et al., 2022; Kobayashi et al., 2022; Yu et al., 2014; Chen et al., 2023). In the LLP binary classification setting, all the training instances are aggregated into *bags* and only the aggregated label count for a bag is available, i.e. proportion of 1's in a bag. Features of all instances are available. This can be seen as a form of weak supervision compared to providing instance-level labels. The main goal is learn an instance wise predictor that performs very well on the test distribution.

There are two sources of information that can help in predicting the instance wise label on the training set. One is the bag level label proportions that are provided. The other source of information is indirect and comes from the fact that any smooth true classifier would likely assign similar labels to instances with similar covariates or feature vectors. Covariate information of instances belonging to the bags are explicitly given. Some of the current methods (Yu et al., 2014; Ardehaly & Culotta, 2017) propose to fit the average soft scores over a bag, predicted by a deep neural network (DNN) classifier, to the given bag label proportion. There is another class of approaches that seek to train a classifier on instance level loss obtained using some form of *pseudo labeling* (Liu et al., 2021; Zhang et al., 2022).

Our work builds on the idea of forming pseudo labels per instance. Our key observation is that one could utilize *two* types of information: *bag level constraints* and *covariate similarity information* in

---

*Equal Contribution. Correspondence to {`shreyasjh`,`navoditasharma`}@google.com

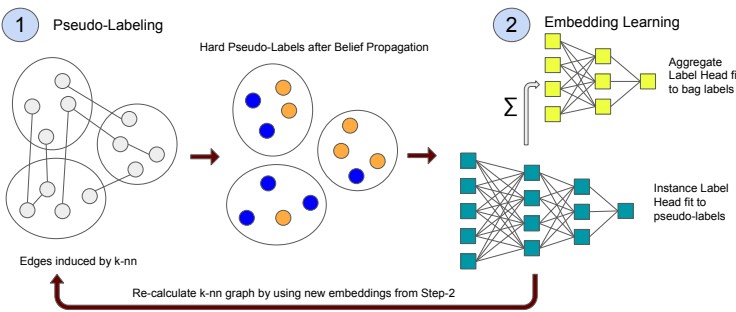

Figure 1: 12 instances placed equally into 3 bags. Step-1: On the k-nn graph induced by the covariate embeddings we perform belief propagation to obtain pseudo-labels that respect edge constraints and bag constraints. Then in Step-2 we fit a MLP to the instance pseudo-labels and bag aggregate label. Embedding learned in an intermediate layer is used to further refine the k-nn graph in Step-1. (Figures best viewed in colour)

an explicit way. To realize this, we take inspiration from coding theory for communication systems (MacKay, 2003; Kschischang et al., 2001), where one of the fundamental problems is to decode an unknown message string sent by the encoder using only parity checks over groups of bits from the message. Parity checks provide *redundancy* that battles against noise/corruptions in the channel. The state of the art codes (MacKay, 2003) are decoded using Belief Propagation (Pearl, 1988) on a factor graph where message bits are variable nodes (whose "label" is to be learnt) and parity checks form factor nodes (that constrain the sum of bits in the parity checks to be odd or even). Sum-product belief propagation is used to learn the marginal soft score on the each of the message bits.

Motivated by the above connection, we propose a novel iterative algorithm that has two stages: We use an existing embedding to learn pseudo-labels and then use the learnt pseudo labels to refine embeddings and we iterate this procedure with new embeddings. Our central approach is described in Figure 1. We outline our contributions in detail below:

1) We draw a parallel between the LLP problem and the parity recovery problem for *pseudo labeling*. The labels of instances are bits of the message and parity checks are the bag level counts (more general than just parity). We adapt this analogy naturally to form a *Gibbs distribution* that enforces the bag constraints. To obtain further redundancy, we exploit covariate information, where, for every pair of instances that are in the $K$-*nearest neighborhood* of each other, we force their binary *labels* to be *similar*, i.e. another "*parity constraint*" is added to the Gibbs distribution. As can be seen in a cartoon depiction in step 1 of Fig. 1, the nearest neighbors induce similar labels (colors). Then, we do a *sum product* Belief Propagation (*BP*) to obtain marginal pseudo labels. To the best of our knowledge, ours is the first work at making this connection between the information theoretic approach of recovering messages from parity and the LLP problem.

2) Our novelty in the second stage is to utilize thresholded soft pseudo labels to provide full supervision to learn a new embedding. However, we observe that after marginalization thresholded soft labels may violate bag constraints. Therefore, we use a novel Deep Neural Network architecture that has an 1) *instance head* giving rise to an instance level loss involving the thresholded soft pseudo label and 2) *bag level head* formed by pooling the penultimate layers's representations of instances within a bag giving rise to a loss between bag proportions and predicted bag level proportion. We call the final loss as *Aggregate Embedding loss*. This loss is used to train the penultimate layer embedding.

3) We iterate the above two steps by using the embedding obtained in the previous step as features. We show that our iterative two stage algorithm, finally produces instance level predictions that outperform a number of LLP baselines including DLLP (Ardehaly & Culotta, 2017) by wide margins. Improvements obtained are upto **15%** on standard **UCI** classification datasets. We outperform the baselines by upto **0.8**% on a large, challenging **Criteo** dataset and upto **7%** on **image** datasets. Our methods mostly outperform most baselines in the large bag regime (upto **15%** gains for bag size $\geq 512$) where supervision is very weak. It is worth noting that on bag size of 2048 there are only a maximum of ~20 bags on the datasets and yet our method does not display significant degradation in performance. Our ablations show that *remarkably* 1-NN achieves most of the gains relative to using $k$-NNs in the pseudo labeling step (Section A.1.1). We find that reducing percentage of KNN constraints (to $50\%$) registers a significant drop in test metrics underscoring the importance of 'parity checks' from covariate nearness (Section 6.2) We provide many other ablations that validate different components of our approach (Section 6 and Section A.1).

## 2 RELATED WORK

**Learning from Label Proportion (LLP):** Quadrianto et al. (2008) use kernel methods under the assumption of class conditioned independence of bags. Yu et al. (2013) were one of the first to tackle the problem and present theoretically backed ∝SVM, a non-convex integer programming solution. This is computationally infeasible on large sized datasets we use. Patrini et al. (2014) introduced a fast learning algorithm that estimates the mean operator using a manifold regularizer while providing guarantees on the approximation bounds. Scott & Zhang (2020) provide an approach that use Mutual Contamination Models which provides some form of weak supervision. The method does not scale to large number of training instances. Several recent methods have been introduced to learn from bagged data. Poyiadzi et al. (2018) propose an algorithm that uses label propagation, i.e. iterative damped averaging of neighbors labels where neighbors are decided based on some similarity measure. Every node is set to the proportion of ones from the bags they participate in at the beginning. This is closest in spirit to ours. However, we always impose bag level constraints and covariate information through a joint Gibbs distribution and use BP to marginalize it followed by an embedding refining step. Poyiadzi et al. (2018) provide results on Size-3 Bags and Size-100 Datasets and their algorithm requires extensive compute for inversion of a kernel matrix. Ardehaly & Culotta (2017) use deep neural networks to tackle the LLP problem show very good empirical performance. When bag size is large their method degrades significantly. Tsai & Lin (2020) use covariate information in the form of a consistency regularizer that modifies the decision boundary. The work does not perform well at larger bag sizes and is computationally expensive for large datasets. Zhang et al. (2022) use forward correction loss to draw parallels to learning from label noise. The method does not converge well for smaller bags. Busa-Fekete et al. (2023) is a very recent work approaching the problem by providing a surrogate loss. We further elaborate on our baselines in section A.4.

Belief propagation in decoding error correcting codes has a long history. There is a related problem of learning from pooled data which has a group testing flavor. We review these in the supplement.

## 3 PROBLEM SETTING AND OVERVIEW OF OUR SOLUTION

Consider a supervised learning dataset $\mathcal{D} = \{(\mathbf{x}_i, y_i)_{i=1}^m\}$ where $x_i \in \mathbb{R}^d$, $y_i \in \{0, 1\}$. The instance wise labels $y_i$'s are not explicitly revealed to the learning agent. There are a set of "bags" $\mathcal{B} = \{S_1 \ldots S_n\}$ that contains subsets $S_i \subseteq [m]$ (thus denoting the indices that correspond to the instances present in that "bag") and for each $S_i$, bag level counts $y(S_i) = \sum_{j \in S_i} y_j^{true}$. In this work, we will consider the case of disjoint bags, i.e. $S_i \cap S_j = \emptyset$. Let the vector of bag level counts be $[y(S_1) \ldots y(S_n)] = y(\mathcal{B})$. In this work, we consider the following problem:

*Given the covariates ($\{x_i\}_{i=1}^m$), information about bag compositions ($\mathcal{B}$) and bag level counts of labels ($y(\mathcal{B})$), our aim is to learn a classifier $f : \mathbb{R}^d \to \{0, 1\}$, $f \in \mathcal{F}$ such that the loss $\ell(\cdot)$ on the test distribution $\mathbb{E}_{(x,y)\sim\mathbb{P}}[\ell(f(x), y)]$ is minimized. Here, $\mathcal{F}$ is the set of classifiers we would like to optimize over. We term this as learning from label proportions problem.*

Our main contribution to this above problem is an iterative procedure that repeats two steps: a) Pseudo Labeling obtained through Sum-Product Belief Propagation that uses a Gibbs distribution capturing covariate information and bag level constraints and b) Learning a better embedding that uses training signals from an instance level predictor on top of the embedding that fits pseudo labels while using the same embedding over multiple instances to simultaneously predict bag level proportions. Then, in the next iteration we use the embedding learnt in the second step instead of original covariates and perform the two steps again. In the last iteration, we simply obtain an instance level predictor to score on the test dataset. Our complete approach is given in Algorithm 1.

## 4 DETAILS OF OUR ALGORITHM

### 4.1 STEP 1: OBTAINING PSEUDO-LABELS THROUGH BELIEF PROPAGATION (BP)

Now, we describe the first step that involves obtaining soft labels from the bag constraints and covariate information. To this end, we form a Gibbs distribution whose energy function captures two-fold information: a) bag level constraints and b) label similarity when two points that are nearby.

---

**Algorithm 1** Iterative Embedding Refinement with BP and Aggregate Embedding

---

**Input:** Covariates $\mathcal{D} = \{x_i\}$, Bag Information $\mathcal{B} = \{S_1, S_2 \ldots S_n\}$, Bag Label Counts $\{Y(S)\}_{S \in \mathcal{B}}$. Parameters $k, \lambda_s, \lambda_b, T, L, L', \tau, \delta_d, k(\cdot, \cdot), d(\cdot, \cdot)$
**Output:** Soft Classifier $f_L(\cdot)$
Set Covariates $\{z_i\} \leftarrow \{x_i\}$
**for** $r \leftarrow 0$ **to** $R$ **do**
  (Pseudo Labeling Step)
  Initialize node and Pairwise potentials $h_i, J_{i,j}$ for Belief Propagation as in equation 3 using $\{z_i\}$.
  $\{\mathbb{P}^r(y_i)\} \leftarrow$ SUM-PRODUCT-BP$(\{h_i\}, \{J_{ij}\})$.
  Obtain Hard Labels: $y_i^r \leftarrow \mathbf{1}_{\mathbb{P}^r(y_i) > \tau}$ .

  (Train embedding with instance and bag losses)
  Train the DNNs $f_L(\cdot), g_L(\cdot)$ (instance and bag loss heads) using hard labels $\{y_i^r\}$ and bag level labels using $\mathcal{L}_{\text{Agg-Emb}}$ loss as in equation 9.

  Set Covariates $\{z_i\} \leftarrow \{f_{L'}(x_i)\}$ (Update Embedding)
**end for**
**Output:** Return the function $f_L(\cdot)$. (Obtain instance label predictor)

---

**Gibbs Distribution from Bag level constraints and covariate information:** The bag level constraints are penalized using a least squares loss between sum of all labels in the bag and the count given by $(\sum_{j \in S_i} y_j - y(S_i))^2$. When two points are close in some distance measure, we would like to make sure their labels are close. In order to capture this, for every point $x_i$ we form $k$-nearest neighbor set $N_k(x_i)$ with respect to a given distance function $d(x, x')$, with $x_j$ added to $N_k(x_i)$ if $d(x_j, x_i) \leq \delta_d$. If $x_j \in N_k(x_i)$, we use the least squares penalization $(y_i - y_j)^2$. We also use a kernel function $k(x, x')$ that scales the least square penalization due to nearness. In most of experiments, we fix $d(x, x')$ to be the cosine distance or euclidean distance. Choices of $k(\cdot)$ are Matern Kernel and RBF kernels.

We define the Gibbs distribution below for the entire dataset $\mathcal{D}$ given by:

$$\mathbb{P}_{\lambda_b, \lambda_s}(y_1..y_m) \propto \exp \left( -\lambda_b \sum_{S_i \in \mathcal{B}} (\sum_{j \in S_i} y_j - y(S_i))^2 - \lambda_s \sum_{x_i, x_j \in \mathcal{D}, x_j \in N_k(x_i)} k(x_i, x_j)(y_i - y_j)^2 \right) \tag{1}$$

This is an Ising model with pairwise potentials and node potentials (external field). We note that both the terms are invariant upto a shift in $y_i$ by a constant. Therefore, transformation to $\{+1, -1\}$ through $y_i' = 2y_i - 1$ would only scale $\lambda_b, \lambda_s$ by a factor of 4. Therefore, the energy function in terms of $\{+1, -1\}$ upto a universal scaling is identical to that of $\{0, 1\}$. Hence, we will remain in $\{0, 1\}$ and state the pairwise and node potentials.

Observing that $y_i^2 = y_i$ and the fact that constant terms in the energy function would not affect the distribution (due to normalization), we can rewrite equation 1 as:

$$\mathbb{P}_{\lambda_b, \lambda_s}(y_1 \ldots y_m) \propto \exp \left( \sum_{i \in [m]} y_i \left[ \sum_{S \in \mathcal{B} : i \in S} \lambda_b(2y(S) - 1) - \lambda_s \sum_{x \in N_k(x_i)} k(x_i, x) \right] + \right.$$

$$\left. \sum_{i \neq j} y_i y_j \left[ 2\lambda_s k(x_i, x_j)(\mathbf{1}_{x_j \in N_k(x_i)} + \mathbf{1}_{x_i \in N_k(x_j)}) - 2\lambda_b |S \in \mathcal{B} : (i, j) \in S|] \right) \right. \tag{2}$$

**Remark:** $|\cdot|$ denotes size of the set satsifying the condition. We note that not all pairwise terms are present. If two points are not in K-NN neighborhood of each other and if they don't belong together in any bag, then there would be no pairwise term corresponding to it. In our experiments, we consider the case where Bags are disjoint and non overlapping. Therefore, every instance $i$ participates in only one bag. Use of K-NN and small disjoint bags creates only linear number of terms in the Gibbs distribution.

**Pairwise and Node potentials:** This is an Ising model $\mathbb{P}(\mathbf{y}) \propto \exp(\sum y_i h_i + \sum_{i \neq j} y_i y_j J_{i,j})$ with node potentials and pairwise potentials given by:

$$h_i = \sum_{S \in \mathcal{B}: i \in S} \lambda_b (2y(S) - 1) - \lambda_s \sum_{x \in N_k(x_i)} k(x_i, x) \tag{3}$$

$$J_{i,j} = 2\lambda_s k(x_i, x_j)(\mathbf{1}_{x_j \in N_k(x_i)} + \mathbf{1}_{x_i \in N_k(x_j)}) - 2\lambda_b |S \in \mathcal{B} : (i, j) \in S| \tag{4}$$

**Obtaining Pseudo Labels using sum-product Belief Propagation (BP):** We use the classical sum-product Belief Propagation (MacKay, 2003) on equation 2 to approximate the marginal distribution $\mathbb{P}_{\lambda_s, \lambda_b}(y_i)$. We briefly describe the algorithm, At every round $t$, node $i$ passes the following message $m_{j \to i}^t(y_i)$, $y_i \in \{0, 1\}$ to every node $j : J_{i,j} \neq 0$ given by:

$$m_{j \to i}^t(y_i) = \sum_{y_j \in \{0,1\}} \exp(y_i h_i) \exp(J_{i,j} y_i y_j) \prod_{k \neq i: J_{k,j} \neq 0} m_{k \to j}^{t-1}(y_j) \tag{5}$$

Here, $m^{t-1}(\cdot)$ represents the message passed in the previous iteration. After $T$ rounds of message passing, we marginalize by using the following (and further normalizing it):

$$\mathbb{P}(y_i) \propto \exp(y_i h_i) \prod_{j: J_{i,j} \neq 0} m_{j \to i}(y_i) \tag{6}$$

**Implementation:** We denote `SUM-PRODUCT-BP`$(\{J_{i,j}\}, \{h_i\})$ to be the sum product belief propagation function that implements $T$ rounds of equation 5 and equation 6. We use `PGMax` package (Zhou et al., 2022) implemented in `JAX` (Bradbury et al., 2018) where we just need to specify the potentials $J_{i,j}$ and $h_i$.

## 4.2 STEP 2: EMBEDDING REFINEMENT LEVERAGING PSEUDO LABELS

We observe that equation 1 uses covariate information by exploiting nearness using nearest neighbors induced by a distance function $d(x, x')$ and using a kernel $k(x, x')$. We now provide an iterative method to refine representation $x_i$ such that points with true labels are brought together progressively although only bag level labels are available. We start with the original features $\{x_i\}$ given to the algorithm (this could already be an embedding obtained from another self-supervised module or any other unsupervised training method like auto encoding).

**Learn marginal Pseudo Labels:** We first identify pseudo labels $\mathbb{P}_{\lambda_s, \lambda_b}(y_i)$ by applying `SUM-PRODUCT-BP` on $h_i, J_{ij}$ obtained from $\{x_i\}$'s. We expect the pseudo labels not to be perfect since it operates on only bag level information and covariate similarity information.

**Learning Embedding using Pseudo Labels:** Let us consider a deep neural net (DNN) classifier (with $L$ layers) of the form given below:

$$f_L(x) = \text{softmax}(W_L^T \sigma_{L-1}(W_{L-1} \sigma_{L-2}(\cdots \sigma_1(W_1^T x + b_1)) + b_{L-1}) + b_L) \tag{7}$$

where $\sigma_\ell$ represents a coordinate wise non-linearity (like `Relu` function), $W_\ell \in \mathbb{R}^{d_\ell \times d_{\ell-1}}$, $\ell \in [1 : L]$ represents a weight matrix at the $\ell$-th layer that multiplies the representation from the $\ell - 1$ layer of dimension $d_{\ell-1}$ and $b_\ell$ represents the biases added coordinate wise to the output which is $d_\ell$ dimensional. In our work, we focus on binary classification where $d_L = 1$ and $d_0 = d$ (input feature dimension). We call $f_L(\cdot)$ the *instance loss* head.

One option is to just fit $f_L(x_i)$ to the information from Pseudo labels $\mathbb{P}_{\lambda_s, \lambda_b}(y_i)$ at the instance level. However, it may not be consistent with the bag level labels in the expected sense. So we impose bag level constraints by first average pooling third to last layer output $f_{L-2}(x)$ across instances $x \in S$ where $S$ is a bag of instances. Then, we have one more hidden layer on top of this pooled representation to finally produce a soft score for the bag proportion. We call this the *bag loss* head and it produces the following soft score for a bag of instances $S \subseteq \mathcal{B}$:

$$g_L(S) = \sigma_L \left( V_L^T \left( \sigma_{L-1} \left( V_{L-1}^T \text{Avg Pooling}[\{f_{L-2}(x_i)\}_{i \in S}] + \tilde{b}_{L-1} \right) + \tilde{b}_L \right) \right) \tag{8}$$

We define two loss functions that learns $f_{L-2}(z)$ representation to simultaneously be consistent with 1) the bag label proportion through the average pooling operation in equation 8 and 2) the other

one that makes it consistent with hard labels obtained by *thresholding* soft pseudo labels given by $y_i^0 = \mathbf{1}_{\mathbb{P}_{\lambda_s,\lambda_b}(y_i)>\tau}$, where $\mathbf{1}_E$ is the indicator function when event $E$ holds.

The composite loss function is given by:

$$\mathcal{L}_{\text{Agg}-\text{Emb}}(S) = \sum_{i \in S} \text{CE}\left(f_L(x_i), \{y_i^0\}\right) + \lambda_a \text{CE}\left(g_L(S), \frac{y(S)}{|S|}\right) \tag{9}$$

Here, CE is the cross entropy loss. We call this composite loss function $\mathcal{L}_{\text{Agg}-\text{Emb}}$ the *aggregate embedding loss* function.

### 4.3 ITERATIVE REFINEMENT

We take the representation computed at some layer $L' < L$ (denoted by $f_{L'}(x)$) and apply the belief propagation (SUM-PRODUCT-BP) again where covariates are given by $\{z_i = f_{L'}(x_i)\}$. We typically use $L' = L - 2$. Then, let the new pseudo labels obtained be $\mathbb{P}^1_{\lambda_s,\lambda_b}(y_i)$ using $\{z_i\}$ as covariates. We obtain hard labels from thresholding pseudo labels by $\tau$ to obtain $y_i^1 = \mathbf{1}_{\mathbb{P}^1_{\lambda_s,\lambda_b}(y_i)>\tau}$

We again fit similar DNNs ($f_L$, $g_L$) by using $\mathcal{L}_{\text{Agg}-\text{Emb}}$ to the new hard labels $\{y_0^1\}$ and the bag level labels $\{y(S)\}$. In principle, we could iterate it several times to refine embeddings progressively but we stop when the new iteration does not clearly improve performance on validation set. Refer to the section A.2.2 for analysis on convergence of our method in 1-2 iterations. When we test on instances, we always remove the bag level head $g_L(\cdot)$ and test it with just the soft score $f_L(x)$. We describe the iterative procedure in Algorithm 1.

## 5 EXPERIMENTS

We perform extensive experimentation on four datasets. We follow the standard procedure of creating disjoint *random* bags where we sample instances without replacement from the training set, and keep repeating this for each bag, *bag-size: k* number of times.

**1. Adult Income** (Dua & Graff, 2017) (Kohavi et al., 1996): Classification task is to predict whether a person makes over $50K a year based on the provided census data of 14 features. The dataset is split 90-10 as train-test and 10% of train is used as a hold out validation following Yoon et al. (2020)

**2. Bank Marketing** (Dua & Graff, 2017) (Moro et al., 2011): The task here it to predict if the client will subscribe a term deposit from 16 features. Data is split as $\frac{2}{3}$-$\frac{1}{3}$ train-test split. We further use $\frac{1}{3}$ of the training set as a hold-out validation set.

**3. Criteo** (Krizhevsky, 2009): 1 week of ad click data to predict CTR with 39 features. We sample non-overlapping sets of 1 million, 200k and 250k instances to form train, validation and test datasets. Note that Criteo is a very challenging benchmark with only +2% AUC improvement shown in the last 7 years (cri, 2023)

**4. CIFAR-10** (Jean-Baptiste Tien, 2014) 60K images with 10 classes. **CIFAR-B**: We assign label *1* to all *Machine* classes (0,1,8,9) and label *0* to all *Animal* classes (2,3,4,5,6,7). In this dataset, 40% of all instances belong to the positive class. **CIFAR-S**: All instances belonging to the class *Ship* are assigned label *1* and all other instances are assigned label *0*. This dataset has 10% positive instances. We use the standard Train-Test splits for these 2 datasets. We use 10% of the data from the Train Set as Validation Set to tune our hyperparameters.

We compare our method against several top LLP Baselines; namely DLLP (Ardehaly & Culotta, 2017), EasyLLP (Busa-Fekete et al., 2023), GenBags (Saket et al., 2022), LLP-FC (Zhang et al., 2022) and LLP-VAT (Tsai & Lin, 2020) described in detail in appendix section A.4

### 5.1 EXPERIMENTAL SETUP

We optimize our algorithm, using hyperparameters $\lambda_s, \lambda_b \in [10^{-4}, 200]$, $k \in [1, 30]$, $T = [50, 100, 200]$, $\tau \in (0, 1)$, $MLP_{LR} \in [10^{-6}, 1]$, $MLP_{WD} \in [10^{-12}, 10^{-1}]$, $\lambda_a \in [0, 10]$, $\delta_d \in [10^{-4}, 1]$, $BatchSize_{train} = [2, 4, 8, \ldots 4096, 8192]$, tuned using Vizier (Song et al., 2022) to achieve the best Validation AUC score. Illustrative values of best hyperparameters for various experiments are given in appendix section A.6. We then report the corresponding Test AUROC %

Table 1: Performance (Test AUROC) on UCI Tabular Datasets on Bag Sizes 8, 32, 128, 512, 1024, 2048 against major baselines. Instance-MLP performance on Adult is **90.30** (0.08) and on Marketing is **86.62** (0.06)

| Bag Size: | 8 | 32 | 128 | 512 | 1024 | 2048 |
|---|---|---|---|---|---|---|
| Dataset: | | | Adult | | | |
| DLLP | 89.19 (0.32) | 87.52 (0.46) | 85.87 (0.91) | 82.95 (1.37) | 63.48 (2.41) | 62.58 (2.18) |
| EasyLLP | 88.68 (0.48) | 87.51 (0.76) | 75.59 (0.84) | 66.02 (1.72) | 61.65 (1.01) | 63.21 (3.53) |
| GenBags | 89.22 (0.32) | 87.72 (0.34) | 86.43 (0.28) | 84.00 (0.26) | 83.52 (0.91) | 80.07 (0.90) |
| Ours-Itr-1 | 89.32 (0.26) | 87.75 (0.32) | 86.70 (0.31) | **84.97 (0.43)** | 83.61 (0.49) | 84.69 (0.76) |
| Ours-Itr-2 | **89.47 (0.29)** | **87.82 (0.33)** | **86.87 (0.39)** | 84.01 (0.34) | **83.88 (0.55)** | **84.95 (0.69)** |
| Dataset: | | | Marketing | | | |
| DLLP | 84.49 (0.70) | 82.65 (0.94) | 79.69 (2.03) | 70.36 (0.64) | 66.39 (2.43) | 65.60 (3.21) |
| EasyLLP | 83.63 (0.34) | 82.87 (0.72) | 75.05 (3.29) | 68.97 (2.76) | 50.23 (1.21) | 50.12 (0.55) |
| GenBags | 85.26 (0.42) | 83.15 (0.34) | 79.74 (0.50) | 69.29 (0.92) | 64.82 (3.10) | 58.43 (4.31) |
| Ours-Itr-1 | 85.76 (0.26) | 84.18 (0.33) | **82.71 (0.44)** | 77.71 (0.46) | 80.56 (0.55) | 78.63 (0.83) |
| Ours-Itr-2 | **86.26 (0.31)** | **84.23 (0.45)** | 82.46 (0.35) | **81.68 (0.77)** | **81.66 (0.61)** | **81.01 (0.92)** |

averaged over 3 trials and report the sample standard deviation in parenthesis throughout our tables. We perform the same setup for all our baselines. Best number is reported in **bold** and 2nd best is reported in underline.

We also run the same MLP on true instance labels. This provides an upper bound to the performance that we can reach using aggregated labels. We report this number for each dataset at the table heading. We use an MLP with 5 Hidden Layers with relu activation and the following number of hidden units: $[5040, 1280, 320, 128, 64]$ for our 2nd Step. The final layer has sigmoid activation. We use Adam optimizer and Binary Cross Entropy Loss for all our datasets. We use the same MLP for all relevant baselines as well. We report main results using $k(\cdot, \cdot) =$ Matern and $d(\cdot, \cdot) =$ Cosine.

We perform experimentation on 6 bag sizes: 8, 32, 128, 512, 1024, 2048. All experiments were performed on a single NVIDIA V100 GPU. We provide further implementation details and the experimental details for the baselines in the supplementary section A.5

## 5.2 PERFORMANCE ANALYSIS

We compare the performance of our method with several baselines on all the datasets. **Ours-Itr-*n*** refers to our method run for *n* iterations.

We use the original features as is for the 2 UCI Datsets with 14 features for Adult Dataset, and 16 features for Marketing Dataset respectively.

Table 1 shows the performance of our method on the two UCI datasets across 6 bag sizes. We make four observations. First, for lower bag size (8,32) our method almost bridges the gap with the instance level performance. Second, the second iteration of our method almost always improves performance over the first iteration across both datasets. Third, we are able to consistently outperform all the baselines across all bag sizes in both datasets. Finally, in large bag regime, our methods perform even better. For instance, with bag sizes 1024 and 2048, we are close to **15%** better compared to the nearest baseline DLLP.

In Table 2 we compare our method with other baselines on the Criteo dataset.[1] We use the self-supervised method MET (Majmundar et al., 2022) to generate embeddings for Criteo dataset to obtain better initial embeddings.This is because most of the features in Criteo are categorical and some of them have large number of categories rendering naive one-hot encoding very intractable. For fairness and consistency we use MET embeddings as input for all the baselines we compare against as well.

Our method scales really well for 1 million samples with negligible computational cost for the sum-product BP step. Criteo is an inherently harder dataset to work with due to high feature dimensionality and most of them being categorical. Over the last 7 years, the dataset has seen a 2% improvement in AUC while our method is able to produce a **0.8%** improvement over DLLP. Similar to the previous tabular datasets, we also observe here that the iteration seems to help improve performance.

---

[1]We were not able to run BP on Criteo for large bag sizes since we ran into integer-overflow issues. It will take some time and perhaps even involved changes to the underlying PGMax library code to resolve them to accommodate large number of factors

Table 2: Performance (Test AUROC scores) on Criteo-1M on Bag Sizes 8, 32, 128 against major baselines. Instance-MLP performance on Criteo is **75.86** (0.04)

| Bag Size: | 8 | 32 | 128 |
|---|---|---|---|
| Dataset: | | Criteo | |
| DLLP | 74.11 (0.09) | 72.86 (0.01) | **70.99** (0.01) |
| EasyLLP | 70.77 (0.92) | 68.42 (0.62) | 62.87 (1.50) |
| GenBags | 73.34 (0.01) | 71.32 (0.77) | 70.39 (0.46) |
| Ours-Itr-1 | 74.96 (0.23) | 73.36 (0.33) | 70.45 (0.51) |
| Ours-Itr-2 | **74.97** (0.24) | **73.43** (0.61) | 70.81 (0.44) |

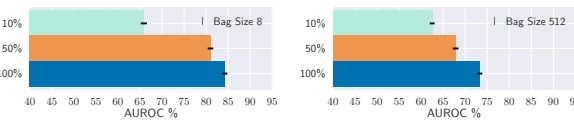

Figure 2: Comparison of the performance on adding different percentage of neighbours on Marketing on bag-size 8 and 512

Table 3: Performance (Test AUROC scores) on Image Datasets on Bag Sizes 8, 32, 128, 512, 1024, 2048 against major baselines. * denotes *SVD did not converge error* (while obtaining noisy labels), ! denotes *Out of Memory Error*. &: Validation performance of first iteration of our algorithm was clearly superior. Instance-MLP AUROC on CIFAR-B is **96.58** (0.04) and on CIFAR-S is **95.06** (0.07)

| Bag Size: | 8 | 32 | 128 | 512 | 1024 | 2048 |
|---|---|---|---|---|---|---|
| Dataset: | | | CIFAR-B | | | |
| DLLP | 95.49 (0.18) | **94.05** (0.09) | **90.90** (0.06) | **86.18** (0.67) | 76.65 (2.62) | 68.19 (1.75) |
| GenBags | 95.32 (0.05) | 93.40 (0.17) | 88.97 (0.33) | 84.30 (0.60) | **82.73** (1.16) | **75.22** (0.75) |
| EasyLLP | 91.33 (0.43) | 84.67 (3.35) | 81.37 (2.28) | 58.07 (13.91) | 65.34 (5.51) | 55.37 (9.62) |
| LLP-FC | 90.19 (0.32) | 88.53 (0.31) | 82.46 (0.49) | 80.12 (1.34) | 78.89 (0.51) | 73.25 (2.26) |
| LLP-VAT | 93.79 (0.29) | 91.38 (0.15) | 88.22 (0.12) | ! | ! | ! |
| Ours-Itr-1 | 95.39 (0.21) | 93.89 (0.18) | 89.28 (0.35) | 85.55 (0.85) | 80.43 (1.51) | 70.06 (1.03) |
| Ours-Itr-2 | **95.54** (0.22) | 93.97 (0.21) | 90.37 (0.27) | 85.92 (0.45) | & | & |
| Dataset: | | | CIFAR-S | | | |
| DLLP | **93.87** (0.11) | **92.12** (0.24) | **88.63** (0.51) | 79.58 (1.34) | 52.01 (8.56) | 57.21 (6.50) |
| GenBags | 92.36 (0.50) | 90.10 (0.39) | 86.78 (0.33) | 82.69 (1.01) | 68.45 (3.79) | 60.43 (4.49) |
| EasyLLP | 85.54 (1.006) | 74.79 (2.17) | 65.26 (3.51) | 61.57 (9.88) | 62.46 (5.21) | 52.32 (3.04) |
| LLP-FC | * | 85.58 (0.31) | 80.59 (0.56) | 75.62 (1.21) | 65.75 (2.36) | 63.76 (1.26) |
| LLP-VAT | 90.10 (0.49) | 83.20 (0.16) | 64.76 (3.06) | ! | ! | ! |
| Ours-Itr-1 | 93.53 (0.39) | 91.29 (0.36) | 88.17 (0.59) | 83.49 (1.53) | **74.45** (2.58) | 71.01 (2.21) |
| Ours-Itr-2 | 93.64 (0.31) | 91.31 (0.33) | 88.31 (0.41) | **84.30** (1.28) | & | **71.17** (2.13) |

In Table 3 we report performance on the CIFAR image dataset. We use the SimCLR (Chen et al., 2020) contrastive learning method to obtain unsupervised embeddings for our experiments on the Image Datasets. We use SimCLR embeddings as input for the relevant baselines we compare. Among the 2 image datasets, CIFAR-S has a large label skew. Our methods outperforms all baseline for CIFAR-S in the large bag size regime ($BagSize \geq 512$), where the performance of all other methods drop significantly. Specifically, we outperform DLLP by upto **20%** and GenBags by upto **7%**. For small bags, our method performs comparable to the best baseline. Only for CIFAR-B that has label balance, GenBags is better than our method for larger bag sizes while DLLP outperforms slightly for lower bag sizes. However, even in this case, our method is competitive (close second mostly) with the best across bag sizes.

**Note:** We justify our chosen hyper-parameters in our experiments via approximate convergence analysis in the supplementary section C providing theoretical backing for our strong empirical results.

## 6 ABLATIONS

Here we provide ablations regarding the two most important ideas in our algorithm: 1) Nearest neighbor based nearness constraints and 2) Aggregate Embedding. Due to space constraints, we do provide a number of other ablations in the appendix regarding adding noise to embeddings (A.1.2), hard vs soft thresholding of BP labels (A.1.3), performance change with different choice of kernels (A.1.5) and distance functions (A.1.4) among others. In the supplement, we further report various performance metrics of our algorithm such as performance of BP pseudo labels in itself compared to ground truth (Section A.2.3) and convergence in very few iterations (typically 1-2) (Section A.2.2).

### 6.1 TIME COMPLEXITY

For the problem we consider in the paper, we have two terms in the BP formulation: KNN based nearness constraints and bag constraints. There are $m/B$ bags each having $B^2$ pairwise terms giving

Table 4: Time for various parts of our algorithm compared to time taken by other methods on Criteo Dataset. All time values are in seconds. Note that *Data Setup time is common* to All Methods

| | Criteo ~1m Samples | | | | | | |
| Bag Size | DLLP Training | EasyLLP Training | GenBags Training | Data Setup | Ours - BP | Ours - MLP | Ours - Total |
|---|---|---|---|---|---|---|---|
| 8 | 725.24 (142.59) | 1617.67 (1597.14) | 1760.33 (1077.05) | 1993.47 (96.05) | 695.34 (266.7) | 679.26 (215.35) | 3368.07 (466.34) |
| 32 | 735.09 (207.50) | 911.86 (553.21) | 957.65 (497.01) | 1970.05 (104.65) | 1279.83 (278.75) | 624.55 (187.30) | 3874.43 (469.92) |
| 128 | 568.00 (79.14) | 777.82 (530.75) | 729.41 (502.12) | 2192.84 (189.90) | 3590.73 (528.85) | 588.00 (157.84) | 6371.57 (727.31) |

rise to $mB$ pairwise terms (here $m$ is the dataset size and $B$ is the bag size). Similar analysis gives $mk$ pairwise terms for the KNN constraints. So we have an Ising Model with $O(m(B+k))$ pairwise terms. Drawn as an undirected graph, the degree is linear in only $B+k$ BP message passing complexity per node per iteration is also $O(B+k)$. Any implementation will only have this much complexity per node per iteration of BP. JAX (Bradbury et al., 2018) implementation in PGMax (Zhou et al., 2022) does an efficient update for all nodes. This is line with increase in complexity of the BP step (Column 3) for Criteo in Table 4. Additional wall clock time results and discussion (that shows linear scaling in bag size for Adult dataset) is in the supplement section A.2.1, Table 5. This establishes the feasibility of our method on larger datasets and larger bag sizes as well.

## 6.2 IMPORTANCE OF NEAREST NEIGHBOR CONSTRAINTS FOR BP

One of the main contributions in our algorithm is the usage of nearness of covariates to impose label similarity constraints in the Gibbs distribution is an unsupervised manner. We show how essential it is by removing a certain percentage of those constraints and studying degradation.

**A good fraction of kNN constraints is necessary:** As depicted in figure 2, we show how for a good fraction of instances neighbourhood information is essential for good performance of the pseudo labeling step using the *Marketing* dataset. By simply retaining only 10% of the pairwise covariate similarity constraints, we lose around **18%** performance compared to when we use the entire set of pairwise covariate factors. This highlights the importance of the covariate information usage in our BP formulation.

## 6.3 LEARNING AGGREGATE EMBEDDINGS HELPS

As we highlight in figure 3, the addition of the additional *bag loss* head in our Aggregate embedding loss in equation 9 pipeline helps improve performance across both smaller and larger bags. Our choice of average pooling of different instances at bag level during the supervised learning provides the best performance. We also note that we experiment with a much more complex choice for aggregation of instances within a bag like using Multi-Head-Attention (*MHA*). This leads to slight degradation in the instance wise performance. We describe the architecture for this choice in the supplement.

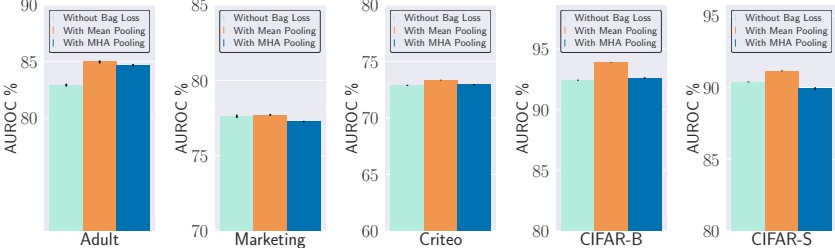

Figure 3: Comparison of the performance on different types of the pooling for the aggregate-embedding loss across different datasets for bag-size 512 for Adult and Marketing and bag-size 32 for Criteo, CIFAR-B and CIFAR-S at the end of iteration 1.

## 7 CONCLUSION

Thus we have provided a highly generalizable algorithm to perform efficient learning from label proportions. We utilised Belief Propagation on parity like constraints derived from covariate information and bag level constraints to obtain pseudo labels. Our unique Aggregate Embedding loss used instance wise pseudo labels and bag level constraints to output a final predictor. We have also provided an theoretical insight into why our approach works through varied ablations on different components and extensive experimental comparisons against several SOTA baselines across various datasets of different types.

ETHICS STATEMENT

The algorithm we propose is implementable over datasets of various kinds, including but not limited to tabular and vision and we demonstrate the efficacy of our approach via comparisons on several such datasets while beating several strong LLP methods. To the best of our knowledge, our work does not raise any ethical concerns.

REPRODUCIBILITY STATEMENT

We have described in detail the implementation details for the reproducibility of the experiments in the main paper Section 5.1 supplementary material Section A.5. We provide the hyperparameter ranges and experimental methodology in section 5.1 and additional information, including that for the baselines in supplementary section A.5. We have provided extensive set of hyperparameters in A.6 to reproduce our algorithm's numbers. We will soon publicly release the source code.

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

# A ADDITIONAL ANALYSIS

## A.1 ADDITIONAL ABLATIONS

### A.1.1 1-NEAREST NEIGHBOR CAPTURES A LOT OF THE GAINS

We now show that using 1 Nearest Neighbour Information for the SUM-PRODUCT-BP is empirically close enough to the best performance we get on the best $k$ chosen for $k$-nearest neighbors used in SUM-PRODUCT-BP by hyperparameter search. From Figure 4, we see that across multiple datasets test metrics are very close for the two settings. This shows lower $k$ which reduces time complexity of the BP step (see section 6.1) does not affect the performance greatly.

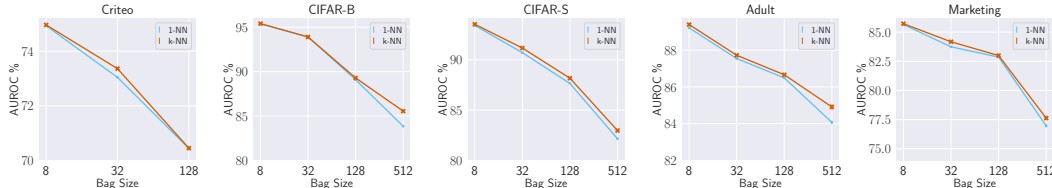

Figure 4: Change in values of Test AUROC when using only 1 nearest neighbour for the covariate factor creation, v/s when using an optimal higher $k$ for the covariate factor creation.

### A.1.2 NOISY EMBEDDINGS AS INPUT

We add noise variables sampled from $\mathcal{N}(0, \sigma^2 I_d)$ to our features input to the first iteration and report the numbers obtained in 2nd-Iteration supervised learning step in figure 5. Medium Noise regime corresponds to $\sigma = 0.05$ and High Noise regime corresponds to $\sigma = 0.1$. As is clearly visible our method is able to recover performance even when using noisy inputs. We would like to note that there is some degradation at bag size level $512$. We point out that this is case where there about $\sim 100$ bag level labels in total. Covariate information is rather crucial to make any progress. Hence noise addition has the most impact in this regime. This also suggests importance of using covariate information for large bag sizes due to very weak supervision available. The drop in performance due to noisy embeddings is significantly higher for the *Marketing* dataset than the *Adult* dataset. This shows that the coariates are very important for the Marketing datasets and our method exploits it very well. We posit thatt his could be the reason our method significantly outperforms others on the *Marketing* dataset (See Table 1).

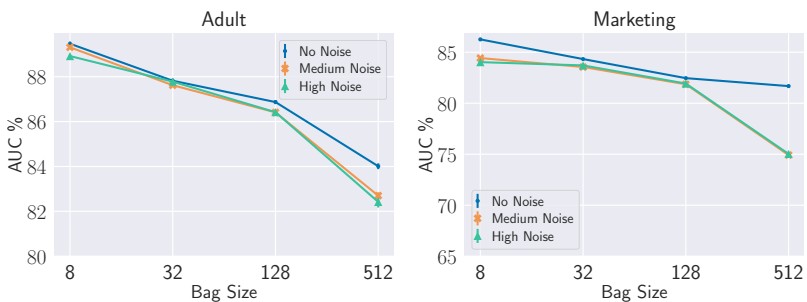

Figure 5: Recovered performance on adding noise to the initial embeddings.

### A.1.3 SOFT WEIGHTED HARD THRESHOLD V/S VANILLA HARD THRESHOLD

In figure 6 we report the numbers obtained on using the soft-labels from the BP-Marginals as opposed to hard thresholding them. We use the soft labels in two ways, directly to train the MLP using a sigmoid cross entropy loss formulation as opposed to the usual binary cross entropy loss, and the other for weighing the hard-labels by $|p - \tau|$ where $p$ is the soft label and $\tau$ the threshold for creating the hard labels. We do not notice any consistent improvements on using the soft labels in either form across datasets and bag sizes and thus stick to hard labels for our setup.

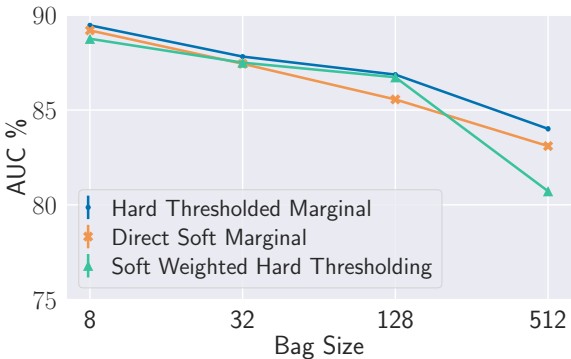

Figure 6: Change in values of MLP AUC when using Soft Weighted Hard Thresholding for the conversion of BP-Marginals to pseudolabels v/s Directly using the soft marginals v/s using Hard Thresholding on Adult Dataset for 2nd Iteration.

### A.1.4 DISTANCE METRIC: COSINE V/S L2

As mentioned earlier, we experiment with using Cosine and L2 distance, $d(\cdot, \cdot)$ for the construction of our neighbour graph. While we don't find significant differences on using the two methods, using Cosine led to better downstream performance across datasets and bag sizes. This can be interpreted to be due to the better neighbour graph construction as depicted in Figure 7 for Adult, by better Test Score (Accuracy) of the kNN constructed by the two distance metrics for varying number of neighbours (k) for the construction of the neighbour graph.

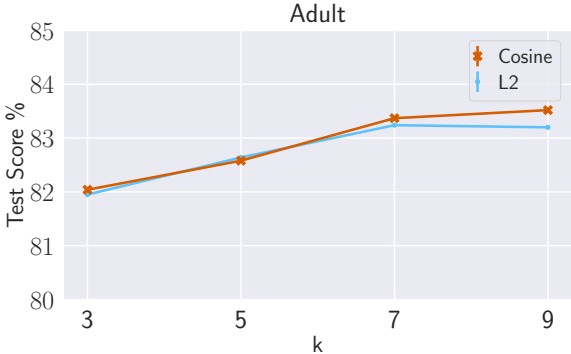

Figure 7: Variation in the kNN Score across bag sizes compared for the two popular distance metrics, Cosine and L2 on Adult Dataset as $d(\cdot, \cdot)$ for the neighbour graph creation.

### A.1.5 SIMILARITY KERNEL: RBF V/S MATERN

As discussed earlier, we tried both the RBF Kernel and Matern Kernel for our experiments and note as in figure 8 for Criteo, that the Matern Kernel resulted in slightly better Test AUC % Scores across various datasets and bag sizes. While there is no substantial increase in our performance the marginally better numbers can be attributed to the fact that the Matern Kernel is a generalization of the RBF Kernel and might capture the similarity information between the embeddings more aptly.

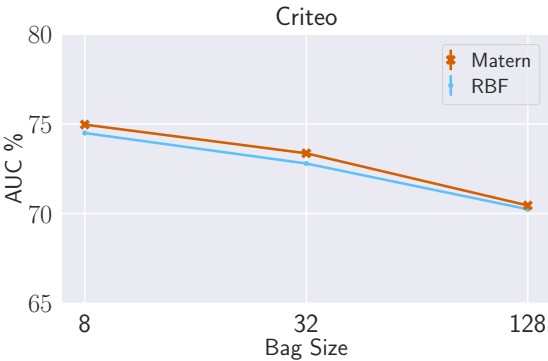

Figure 8: Variation in the Test AUC % of the MLP after 1st Iteration on using RBF Kernel v/s on using Matern Kernel as $k(\cdot, \cdot)$ for similarity calculation during formation of pairwise factors on Criteo.

### A.1.6 OPTIMAL VALUES OF $\lambda_a$

As observed in Figure 9, it is clear that the bag-loss head plays a more important role when the bag sizes are smaller. The optimal value of $\lambda_a$ is dependent on the requirement of the reinforcement of the bag constraint via the bag-loss head. For small bags, the bag constraints hold much more information than that for large bags, and hence are more useful. In the case of small bags, during aggregation of embeddings, more information is retained and utilized downstream since fewer embeddings are pooled. We pool 8 embeddings per bag for bag size 8 and 512 embeddings are pooled for bag size 512, clearly there is a stark divide in the information summarized across bag sizes. This trend is consistently noticed across multiple datasets. We also notice that the optimal values of $\lambda_a$ are comparatively lower for the 2nd Iteration of our method. We think this might be due to the refined embeddings and neighbour graph in the 2nd iteration of Belief Propagation.

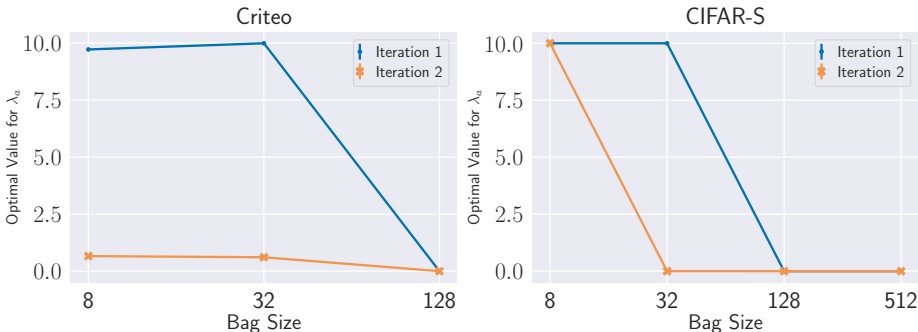

Figure 9: Variation in the Test AUC % of the MLP after 1st Iteration on using RBF Kernel v/s on using Matern Kernel as $k(\cdot, \cdot)$ for similarity calculation during formation of pairwise factors on Criteo.

## A.2 EXTENDED EXPERIMENTATION

### A.2.1 WALL CLOCK TIMES ON ADULT DATASET

We have wall clock time reported in Table 4 (main paper) and Table 5 for Criteo and Adult respectively. Criteo has 1 million samples in total in the training. For bag size 32, the wall clock time of the entire BP stage is only 1279s for millions of factors in the factor graph inclusive of the creation and iteration. For smaller datasets like Adult with 50k samples, even on bag size as large as 2048, the BP

stage takes only 1054s on a NVIDIA P100 GPU, which is a minimal computational overhead above standard supervised learning. It is also conventionally known that BP on sparse graphs is very fast and our sparsity is controlled by using K-NN instead of all pairs in imposing nearness constraints. We reiterate that the number of factors in the factor graph is thus only linear in the number of samples allowing for faster iterations. The time taken in the entire BP section, right from creation of the factor graph to message passing for 100 iterations happens in the order of $O(\text{bag\_size})$ seconds for the Adult dataset which is exceptionally fast as simple MLP training itself takes $O(10^2)$ seconds.

Table 5: Time for various parts of our algorithm compared to time taken by other methods on Adult Dataset. All time values are in seconds. Note that Data Setup time is common to All Methods

| | | | Adult ~50k Samples | | | |
|---|---|---|---|---|---|---|
| Bag Size | DLLP Training | EasyLLP Training | GenBags Training | Data Setup | Ours - BP | Ours - MLP | Ours - Total |
| 8 | 52.23 (51.38) | 47.85 (28.93) | 256.94 (495.04) | 630.88 (827.63) | 12.34 (8.57) | 85.00 (145.57) | 728.21 (863.69) |
| 32 | 45.4 (69.04) | 26.51 (12.67) | 125.67 (102.96) | 523.59 (449.32) | 21.11 (14.65) | 54.38 (77.15) | 599.08 (458.12) |
| 128 | 73.79 (125.96) | 21.38 (15.99) | 88.72 (76.21) | 623.20 (458.23) | 71.16 (8.89) | 54.34 (63.55) | 748.71 (459.86) |
| 512 | 25.11 (40.15) | 18.84 (9.42) | 85.52 (74.49) | 767.58 (431.50) | 269.88 (32.52) | 65.07 (158.34) | 1102.53 (459.12) |
| 1024 | 26.39 (38.06) | 20.93 (14.51) | 95.79 (103.85) | 645.04 (674.94) | 550.32 (113.28) | 32.42 (25.76) | 1227.78 (756.34) |
| 2048 | 24.34 (28.32) | 18.77 (6.69) | 86.14 (71.80) | 612.87 (539.07) | 1054.47 (73.71) | 49.93 (78.34) | 1717.27 (534.88) |

### A.2.2 Convergence of the two step method

In Table 6 empirically we demonstrate that our method converges in two iterations by looking at relative improvements between iterations 2 and 3. We demonstrate that two iterations of our algorithm suffice empirically.

Table 6: Empirical Convergence: The % AUC scores for varying bag sizes for 2 Iterations and 3 Iterations of our algorithm.

| | Adult | | | | Marketing | | | |
|---|---|---|---|---|---|---|---|---|
| Bag Size | 8 | 32 | 128 | 512 | 8 | 32 | 128 | 512 |
| Itr-2 | **89.47** | **87.82** | 86.87 | 84.01 | **86.26** | **84.33** | **82.46** | **81.68** |
| Delta | -0.29 | -1.26 | 0.49 | 0.71 | -0.26 | -0.94 | -0.13 | -1.05 |
| Itr-3 | 89.18 | 86.56 | **87.36** | **84.72** | 86.00 | 83.39 | 82.33 | 80.63 |

As visible, performance gains from 2nd to 3rd iterations are not consistently better. Thus there is no clear reason to run higher iterations of the algorithm, as a maximum of 2 iterations suffice to achieve significantly consistent performance.

### A.2.3 Goodness of Pseudo Label of BP

We report the AUROC of BP after the first iteration with respect to the true labels in Table 7. It is considerable indicating that it has good ordering information (ranking of samples belonging to class 1 above class 0). The effect of high quality pseudo labels is reflected in the downstream performance.

Table 7: The % AUC scores of the pseudo labels obtained from the Belief Propagation algorithm when compared to the ground truth labels for iteration 1 of the algorithm

| Bag Size | 8 | 32 | 128 | 512 | 1024 | 2048 |
|---|---|---|---|---|---|---|
| Adult | 86.34 | 79.63 | 75.25 | 75.02 | 67.88 | 63.54 |
| Marketing | 88.53 | 78.26 | 75.5 | 75.66 | 74.9 | 74.64 |

While Table 7 only reports after Step 1 of iteration 1 to showcase value of the BP step, the second aggregate embedding loss based MLP training Step 2 boosts performance of Step 1 further and we do see it as expected in Table 1. Refer to Section 4, subsection 4.1 for Step 1, and subsection 4.2 for description of these steps in our algorithm.

Step 2 is necessary because information from pseudo labels may not satisfy bag constraints exactly. So we have a composite loss that again imposes the bag constraint through an aggregate embedding loss (see Equation 8 and Equation 9 in the paper in Section 4.2)

### A.2.4 NOISY LABELS AND PRIVACY

In this section we explore utility-privacy tradeoff of our algorithm when we add noise to label proportions for every bag by Gaussian Mechanism to achieve a target *label differential privacy* of $(\epsilon, \delta)$ by using the following result:

**Theorem 1** (Theorem 2 in (Dwork et al., 2014)). *Let $f : \mathcal{A} \to \mathbb{R}$ be a real-valued function. Let $\tau = \Delta f \sqrt{2 \ln(1.25/\delta)}/\epsilon$. The Gaussian Mechanism, which adds independently drawn random noise distributed as $\mathcal{N}(0, \tau^2)$ to output of $f(A)$, ensures $(\epsilon, \delta)$-differential privacy.*

We take $\mathcal{A}$ to the set of true labels of instances in bag $S \in \mathcal{B}$, $f(\cdot) = \frac{y(S)}{B}$, the label proportion. We observe that sensitivity of the label proportion to change in a single label is $\Delta f = \frac{1}{B}$, where $B$ denotes the bag size. Standard deviation of the noise added is proportional to $1/B$ for a fixed $\epsilon, \delta$.

We demonstrate the following interesting privacy-utility tradeoff: utility degradation, as measured by Test AUC, due to Gaussian Mechanism is much more in smaller bags as compared to larger bags for a target privacy level. Through this, we empirically verify the intuition that points to the fact that larger bags offer better privacy. We note that our algorithm performs much better in large bags regime compared to baseline and we conjecture that this is crucial to utilize the better privacy utility degradation tradeoffs at larger bag sizes.

We experiment with 2 sets of differential privacy parameters, Medium Noise: $(\delta, \epsilon) = (10^{-5}, 10)$ and High Noise: $(\delta, \epsilon) = (10^{-5}, 1)$, both popular choices in literature (Papernot et al., 2016) Note that, bag size $B$ takes values in $\{8, 32, 128, 512\}$. Our results are reported in Table 8.

Table 8: Test AUROC scores after Iteration-1 of our method across different bag sizes for varying levels of label-noise.

| Dataset: | Criteo | | | CIFAR-B | | | CIFAR-S | | |
|---|---|---|---|---|---|---|---|---|---|
| Noise: | Noiseless | Medium | High | Noiseless | Medium | High | Noiseless | Medium | High |
| 8 | 74.96 (0.01) | 74.83 (0.07) | 71.06 (0.15) | 95.39 (0.01) | 94.80 (0.04) | 90.99 (0.1) | 93.53 (0.03) | 93.28 (0.28) | 87.07 (0.46) |
| 32 | 73.36 (0.03) | 72.43 (0.02) | 70.40 (0.03) | 93.89 (0.02) | 93.36 (0.05) | 90.25 (0.06) | 91.17 (0.03) | 91.07 (0.21) | 86.08 (0.07) |
| 128 | 70.45 (0.05) | 69.45 (0.20) | 69.53 (0.21) | 89.28 (0.05) | 88.92 (0.13) | 87.79 (0.09) | 88.17 (0.19) | 87.39 (0.10) | 85.17 (0.23) |
| 512 | - | - | - | 85.55 (0.75) | 83.29 (0.32) | 84.65 (0.21) | 82.97 (0.33) | 81.32 (0.37) | 79.39 (0.58) |

We also want to highlight that under the effect of both medium and high noise our method recovers performance up to a reasonable degree, especially for larger bag sizes which as stated earlier are more important from a privacy perspective.

### A.2.5 MAP DECODING

Table 9 and Table 10 denote the Test AUC % on performing Max Product BP on the Gibbs Distribution, and highlight the noisy nature of the performance. This establishes the superiority of Sum Product BP, that is our approach, to obtain consistently good performance across bag sizes and datasets. On the other hand, the single label configuration obtained from the Max Product approach is noisy and does not consistently retrieve the same performance as in the Sum Product approach across bag sizes.

One plausible reason is that given the nature of the weak supervision, i.e. aggregate labels being available only at the level of bags, it is better to find out uncertainty in a label for an instance (marginalize) rather than commit to a MAP configuration with respect to a Gibbs distribution that is uncertain.

Table 9: MaxProduct BP on Adult Dataset

| Bag Size | Test AUC Itr-1 | Test AUC Itr-2 |
|---|---|---|
| 8 | 89.15 | 88.92 |
| 32 | 75.29 | 87.9 |
| 128 | 76.08 | 84.67 |
| 512 | 73.77 | 73.82 |
| 1024 | 80.43 | 76.65 |
| 2048 | 74.45 | 68.69 |

Table 10: MaxProduct BP on Marketing Dataset

| Bag Size | Test AUC Iter 1 | Test AUC Iter 2 |
|---|---|---|
| 8 | 85.55 | 85.73 |
| 32 | 83.99 | 83.67 |
| 128 | 82.64 | 82.39 |
| 512 | 72.29 | 80.66 |
| 1024 | 72.48 | 73.03 |
| 2048 | 72.49 | 80.42 |

### A.2.6 Extension to the Multiclass Paradigm

We adapted the Gibbs measure to the multi class setting as follows. Every point has $k$ labels : $y_i^1 \ldots y_i^k$ corresponding to $k$ classes.

We have three main types of terms in the Gibbs measure. We impose a soft one hot constraint with the term: $(\sum_p y_i^p - 1)^2$

Nearness terms get modified as follows: $K(x_i, x_j) \sum_p (y_i^p - y_j^p)^2$, i.e. Euclidean distance between the vector labels of two points is small if they are nearby.

Aggregate Bag level constraints have counts $b_1 \ldots b_k$. Then we simply impose a least squares constraint: $\sum_p (\sum_{i \in B} y_i^p - b_p)^2$

All these terms are scaled by temperature hyper parameters which we search over during training. Essentially these are vectorized least squares constraints.

The results on CIFAR10 as provided in Table 11 depict that our method is slightly better or comparable to the SOTA.

Table 11: % Accuracy of our method against high performing baselines on CIFAR10 multiclass classification.

| Bag Sizes | 8 | 32 |
|---|---|---|
| LLP-VAT | 66.91 (1.23) | 60.85 (2.45) |
| LLP-FC | 66.52 (2.12) | 62.35 (1.32) |
| DLLP | 68.14 (0.48) | 62.87 (0.88) |
| Ours-Itr-1 | 68.05 (0.23) | 61.83 (0.21) |
| Ours-Itr-2 | 69.23 (0.12) | 61.92 (0.43) |

## A.3 Analysis of 1-NN Graphs

Section A.1.1 shows that running our algorithm with 1-NN for the BP step captures most of the performance in terms of the final Test AUC score. We show that many parts of the factor graph in the case of 1-NN are cycle free.

Consider the bi-partite factor graph $K(V, \mathcal{B} \cup \mathcal{F}, E)$ where variable nodes $V = [1 : N]$ representing $\{x_i\}$ form one partition and bag factor nodes $\mathcal{B}$ and 1-NN factor nodes $\mathcal{F} = \{f : (f, i), (f, j) \in E, \ x_i \in N_1(x_j) \vee x_j \in N_1(x_i)\}$ are on the other partition. Edges between a bag factor node $S$ and variable node $i$ exists if $i \in S$ ($x_i$ belong to bag $S$).

In our setup (experiments), all bag factor nodes have disjoint neighbors since bags are formed randomly without replacement. Therefore, the bi-partite factor graph between $\mathcal{B}$ and $V$ is a forest. Now, consider the bi-partite factor graph between $\mathcal{F}$ and $V$. We now show that this is also a forest. Since every factor node connects only a pair of distinct nodes it is enough to show that the undirected 1-NN graph does not have any cycles.

We now show that the 1-NN graph does not have any undirected cycles. Recall that $N_1(x)$ is the nearest neighbor of point $x$.

**Lemma 1.** *For a set of points $\{x_i\}_{i=1}^N$, consider the following undirected graph $G(V, E)$ where $V = [1 : N]$ and $E = \{(i, j) : x_i \in N_1(x_j) \vee x_j \in N_1(x_i)\}$. For every node $i$, if the edge set is*

*formed by choosing one amongst many equivalent nearest neighbors of $i$ appropriately (i.e. $N_1(x_i)$ is chosen to be a singleton), then $G$ does not have any cycles.*

*Proof.* Let us define a directed graph $G_d$, where the outgoing edge $(i \to j)$ exists if $x_j$ is the closest neighbour of $x_i$ where ties are broken arbitrarily. Thus every node $i$ has out-degree of at-most 1. However, note that the in-degree of any node can be $> 1$. Note that, $G$ can be obtained by replacing oriented edges in $G_d$ by undirected edges. Further, if $(i \to j), (j \to i)$ both exists, we replace it by one undirected edges $(i, j) \in G$.

There are 3 types of cycles in $G_d$:

1. Directed cycle with at least 3 distinct elements (except end point which is repeated). An example of this kind (of length 3) is illustrated in Figure 11.

2. Cycle with a collider whose undirected skeleton forms a cycle in $G$ of length at least 3 with distinct elements. This is illustrated in Fig. 12.

3. Directed cycle $i \to j \to i$. This is illustrated in Figure 10.

Now, we proceed to show that the first two cycles are not possible. Since a bi-directed edge in $G_d$ (Figure 10) will get replaced by a single un-directed edge in $G$, this proves the Lemma.

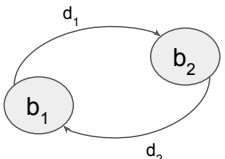

Figure 10: Cycle of Type 3

**Case 1:** We deal with the directed cycle by first proving a claim about a directed path in $G_d$.

**Claim 1.** *Consider a directed path in $G_d$ of the form $a_1 \xrightarrow{d_1} a_2 \xrightarrow{d_2} a_3 \xrightarrow{d_3} a_4 \ldots a_n$ with distinct elements. Here, $d_i$ notes the distance $d(x_{a_i}, x_{a_{i+1}})$. Then, $d_1 \geq d_2 \geq d_3 \ldots d_{n-1}$.*

*Proof.* Let us prove this by contradiction. Say this was not true, then without loss of generality suppose $d_i < d_{i+1}$. We know that the outgoing edge represents the nearest neighbour of a node. If $d_i < d_{i+1}$ was indeed true, then the nearest neighbour of $a_{i+1}$ would have been $a_i$ and not $a_{i+2}$ which is clearly not the case since the edge $a_{i+1} \xrightarrow{d_{i+1}} a_{i+2}$ exists in $G_d$. We get a contradiction and therefore the claim is proven. $\square$

Now, we consider the directed cycle $a_1 \xrightarrow{d_1} a_2 \xrightarrow{d_1} a_3 \ldots a_n \xrightarrow{d_n} a_1$ where all elements $a_i$ are distinct.

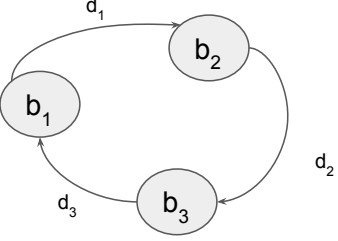

Figure 11: Cycle of Type 1

From, Claim 1 we have $d_1 \geq d_2 \geq d_3 \geq d_n \geq d_1$ applying it on two directed paths $a_n, a_1, a_2$ and $a_1, a_2 \ldots a_n$. This is only possible if $d_1 = d_2 \ldots = d_n$. In this case, one can form an alternate $G_d$ by breaking the cycle where one can have $a_2 \to a_1$ instead of $a_2 \to a_3$.

Therefore, this type of a cycle cannot exist. If it exists, it can be broken by re-assigning an equivalent nearest neighbor.

**Case 2:** Consider a configuration that is a undirected cycle in $G$ but not a directed cycle in $G_d$. Then, it is a cycle consists of paths $a_1, a_2 \ldots \to a_n$, $a_1, b_2, b_{n-1} \to a_n$ where they *collide* at $a_n$. Here all nodes $a_i, b_i$ are distinct. An Example is the collider $c_3$ in Fig. 12. Other orientations are left unspecified in this cycle. For this to there must exist $a_i \neq a_n$ or $b_i$ that has 2 outward edges in this cycle. However, $\mathrm{out} - \mathrm{degree} \geq 2$ is clearly cannot be possible as we are only dealing with 1-NNs and each node can have at-most one outward edge. Therefore such a cycle is not possible. In the Figure 12, the edge $c_1 \to c_4$ or $c_1 \to c_2$ cannot exist as $G_d$ is a directed 1-NN graph and thus such a cycle cannot exist.

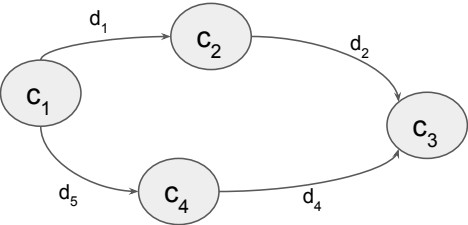

Figure 12: Cycle of Type 2

Thus, we have shown that no cycle can exist in the neighbour graph $G$. □

## A.4 BASELINES

We compare our methods with the following baselines.

**1. DLLP**: We use the DLLP method from Ardehaly & Culotta (2017) as a baseline for both Tabular and Image Datasets. This method fits the prediction score averaged over a bag of a deep classifer to bag level proportions.

**2. EasyLLP**: This was proposed in Busa-Fekete et al. (2023) and we use this as a competitive baseline on all our datasets. They define a surrogate loss function based on the global label proportion.

**3. GenBags**: Introduced in Saket et al. (2022) is another popular algorithm for tabular datasets. The algorithm combines bag distributions, if possible, into good generalized bag distributions, which are then trained on by using standard proportion loss.

**4. LLP-FC**: This methods was introduced in Zhang et al. (2022) where LLP problem was reduced to learning from label noise problem. We use this for image datasets on which it was applied in Zhang et al. (2022).

**5. LLP-VAT**: Method from Tsai & Lin (2020) that we use on Image Datasets. This method is inspired by consistency regularization to produce a decision and approach LLP from a semi-supervised angle. boundary that better describes the data manifold

## A.5 IMPLEMENTATION DETAILS

In most of our experiments, we fix $d(x, x')$ to be Cosine Distance: $1 - \frac{x \cdot x'}{\|x\|_2 \|x'\|_2}$ or Euclidean Distance: (Minkowski Distance with p=2) $\left( \sum_{i=1}^{n} |x_i - x_i'|^p \right)^{\frac{1}{p}}$ and we choose $k(x, x')$ to be one of RBF Kernel: $\exp(-\gamma \cdot d(x, x')^2)$ or Matern Kernel: (a generalization of the RBF kernel) $\frac{1}{\Gamma(\nu)2^{\nu-1}} \left( \frac{\sqrt{2\nu}}{l} d(x, x') \right)^{\nu} K_\nu \left( \frac{\sqrt{2\nu}}{l} d(x, x') \right)$ where $d(\cdot, \cdot)$ is the Euclidean distance, $K_v(\cdot)$ is a modified Bessel function and $\Gamma(\cdot)$ is the gamma function. We justify our choice via the slightly

superior performance observed on using Cosine Distance and Matern Kernel as discussed in ablations in A.1.4 and A.1.5 respectively.

The bags are batched into batches of size $max(BatchSize_{train}, TotalBags)$. We always run the second step, the MLP training for max 100 epochs, using Adam Optimizer with $MLP_{LR}, MLP_{WD}$ learning rate and weight decay respectively. We use the Early Stopping criterion to decide when to stop training. According to this, we stop training if the validation AUC does not increase for 20 consecutive epochs, and then restore the model with the best validation AUC. Such a validation-based early stopping technique is quite popular in literature and well described in Prechelt (2002). Attached is the tensorflow callback code that we implement for the same, based on documentation provided in ker (2023):

```
tf.keras.callbacks.EarlyStopping(
        monitor='val_auc',
        patience=20,
        restore_best_weights=True,
        min_delta=0,
        verbose=1,
        mode='max',
    )
```

We use the official GitHub Implementations of Saket et al. (2022), Zhang et al. (2022) and Tsai & Lin (2020) and perform a grid search over relevant mentioned parameters in their readme. We use WideResNet-16-4 (Zagoruyko & Komodakis, 2016) as the backbone for LLP-VAT and LLP-FC methods as it provides the best performance. For methods described in Busa-Fekete et al. (2023), Ardehaly & Culotta (2017) we implement the algorithm described in the paper with the same MLP as described in 5.1 and sweep the appropriate hyperparams as described in the respective papers.

For Pooling with MultiHeadAttention we use the standard MultiHeadAttention framework from Set Transformers (Lee et al., 2019) using $d = 128$ dimensional embeddings as input (the 2nd last hidden layer of our MLP), 2 heads, 1 seed vector, and 2 row-wise feedforward layers each of size $d$.

## A.6 ILLUSTRATIVE BEST HYPER-PARAMETER VALUES

Here we provide the set of hyperparameters to reproduce the numbers obtained for the first iteration of our algorithm across all datasets and bag sizes in Table 12, Table 13, Table 14, Table 15 and Table 16.

Table 12: The set of hyperparameters for various bag sizes for Adult Dataset for the first iteration.

| Bag Size | $\lambda_s$ | $\lambda_b$ | $\lambda_a$ | $MLP_{LR}$ | $MLP_{WD}$ | $k$ | $\tau$ | $\delta_d$ | T |
|---|---|---|---|---|---|---|---|---|---|
| 2048 | 0.0001 | 0.0184 | 0.0001 | 0.0007 | 1.00E-12 | 1 | 0.03558 | 1 | 100 |
| 1028 | 0.0001 | 0.0576 | 0.0001 | 0.0012 | 1.00E-12 | 1 | 0.01 | 1 | 100 |
| 512 | 0.003 | 0.0796 | 0.0001 | 0.00033 | 0.00025 | 1 | 0.03 | 1 | 100 |
| 125 | 0.0001 | 0.3422 | 10 | 0.00028 | 0.1 | 1 | 0.0216 | 0.01 | 100 |
| 32 | 186 | 0.1659 | 7.5 | 0.00014 | 2.12E-11 | 17 | 0.3327 | 0.6 | 100 |
| 8 | 0.0001 | 0.4427 | 10 | 0.001 | 1.00E-12 | 1 | 0.3515 | 1 | 100 |

Table 13: The set of hyperparameters for various bag sizes for Marketing Dataset for the first iteration.

| Bag Size | $\lambda_s$ | $\lambda_b$ | $\lambda_a$ | $MLP_{LR}$ | $MLP_{WD}$ | $k$ | $\tau$ | $\delta_d$ | T |
|---|---|---|---|---|---|---|---|---|---|
| 2048 | 1.928 | 7.6986 | 0.0136 | 0.0002 | 3.70E-07 | 15 | 0.5311 | 0.1489 | 100 |
| 1028 | 0.0001 | 0.02227 | 0.000167 | 0.00005 | 0.06129 | 28 | 0.03077 | 0.4963 | 100 |
| 512 | 0.00287 | 0.2676 | 0 | 0.00053 | 0.0825 | 7 | 0.0815 | 1 | 100 |
| 125 | 200 | 26.058 | 10 | 0.0027 | 0.0007735 | 28 | 0.03357 | 0.01 | 100 |
| 32 | 200 | 200 | 9.661 | 0.00058 | 4.54E-12 | 29 | 0.0111 | 1 | 100 |
| 8 | 4.146 | 2 | 0.0001 | 0.00085 | 9.20E-11 | 2 | 0.2023 | 1 | 100 |

Table 14: The set of hyperparameters for various bag sizes for Criteo Dataset for the first iteration.

| Bag Size | $\lambda_s$ | $\lambda_b$ | $\lambda_a$ | $MLP_{LR}$ | $MLP_{WD}$ | $k$ | $\tau$ | $\delta_d$ | T |
|---|---|---|---|---|---|---|---|---|---|
| 128 | 0.4359 | 0.2265 | 0 | 0.000001 | 0.0000078 | 11 | 0.14294 | 0.00288 | 200 |
| 32 | 0.0003 | 0.2502 | 9.9885 | 0.00002368 | 0.000547 | 23 | 0.5335 | 0.1326 | 200 |
| 8 | 0.00015 | 0.1884 | 9.716 | 0.00006 | 0.0009567 | 29 | 0.4104 | 0.9996 | 200 |

Table 15: The set of hyperparameters for various bag sizes for CIFAR-S Dataset for the first iteration.

| Bag Size | $\lambda_s$ | $\lambda_b$ | $\lambda_a$ | $MLP_{LR}$ | $MLP_{WD}$ | $k$ | $\tau$ | $\delta_d$ | T |
|---|---|---|---|---|---|---|---|---|---|
| 2048 | 0.057 | 0.02646 | 0.000156 | 0.000589 | 0.0000018 | 3 | 0.02687 | 0.6339 | 200 |
| 1024 | 0.00015 | 0.0175 | 0.00169 | 0.0006 | 0.000025 | 15 | 0.1214 | 0.9697 | 200 |
| 512 | 34.32 | 0.0338 | 0 | 0.00038 | 0.000014 | 1 | 0.267 | 0.2614 | 200 |
| 128 | 0.3674 | 0.105 | 0 | 0.0016 | 0.00476 | 9 | 0.31 | 0.01823 | 200 |
| 32 | 12.43 | 1.2541 | 10 | 0.000068 | 0.000336 | 4 | 0.4934 | 0.3727 | 200 |
| 8 | 0.0001 | 0.8555 | 10 | 0.0002 | 0.00001 | 28 | 0.1961 | 0.4421 | 200 |

Table 16: The set of hyperparameters for various bag sizes for CIFAR-B Dataset for the first iteration.

| Bag Size | $\lambda_s$ | $\lambda_b$ | $\lambda_a$ | $MLP_{LR}$ | $MLP_{WD}$ | $k$ | $\tau$ | $\delta_d$ | T |
|---|---|---|---|---|---|---|---|---|---|
| 2048 | 0.00043 | 0.8279 | 0.0002 | 0.000001 | 0.000001 | 2 | 0.5936 | 0.2771 | 200 |
| 1024 | 0.0001 | 0.003556 | 0.00695 | 0.00032 | 0.000001 | 4 | 0.435 | 0.6544 | 200 |
| 512 | 0.1289 | 0.00754 | 0 | 0.0116 | 0.0011 | 14 | 0.4337 | 0.407 | 200 |
| 128 | 0.0001 | 0.016 | 0 | 0.00214 | 0.00002 | 25 | 0.4508 | 0.0001 | 200 |
| 32 | 0.000192 | 0.0968 | 8.8918 | 0.000096 | 0.000723 | 1 | 0.4294 | 0.00385 | 200 |
| 8 | 0.0008 | 0.099 | 10 | 0.0000013 | 0.000009 | 1 | 0.4856 | 0.2427 | 200 |

## B EXTENDED RELATED WORK

**Belief Propagation**: Belief Propagation (BP) has been used to compute marginals and find MAP estimates in standard sparse graphical models, like Bayesian networks and Markov random fields (Pearl, 2022) by message passing across edges on an appropriate graph. Sum-product BP algorithm is used for computing marginals and it is known to converge on trees. It was also extended to polytrees (Kim & Pearl, 1983). It is also an effective approximate algorithm on general graphical models (Pearl, 1988). More relevant to our work is the fact that sum product Belief Propagation has found widespread in communication system, where it is used to soft-decode a binary string message from their parity checks as in LDPC (low-density parity-check) codes (Richardson & Urbanke, 2001; Gallager, 1962), iterative decoding of turbo codes (MacKay, 2003; Kschischang et al., 2001). In communication codes, parity checks are designed so as to have nice properties on the graphical models they induce. In our problem, the bag levels constraints can be thought of as parity checks but are given and we add additional constraints from covariate information that is also given. We use a public scalabale and efficient implementation PGMax (Zhou et al., 2022) of the sum-product message passing algorithm.

**Decoding from Pooled Data:** Another very relevant area of work learning from pooled data paradigm Scarlett & Cevher (2017); El Alaoui et al. (2018) where the aim to identify the categorical labels of a large collection of items from histogram information at the bag level. El Alaoui et al. (2018) present an approximate message passing algorithm for decoding a discrete signal of categorical variables from several histograms of pooled subsets of data. This line of work is also largely aimed at the regime of very large bags (Scarlett & Cevher, 2017) ($\sim O(\frac{n}{\log n})$) and there is no covariate information available. In our problem, bags are constant in size and they are disjoint.

**Weak Supervision:** There has been recent interest in exploring training of models under weak supervision, where complete label information is not available. Though these methods do not exactly map to the learning from label proportions setup, the idea of using pseudo labels to train a model with pre-trained representations has been explored before in Chen et al. (2022), Pukdee et al. (2022), Ratner et al. (2017), Karamanolakis et al. (2021), which creatively bypass the lack of labels, by either using covariate information, label propagation based on the few available labels, heuristically creating new labels or even using student-teacher models. While such approaches are designed to work in the case of lack of all instance labels, none of them deal with the specific case of weak supervision

we concern ourselves with, namely learning only from the aggregate bag labels and no instance labels whatsoever. This is what makes the LLP setup even harder and sparser in terms of information available for the learner.

## C   Approximate Convergence Analysis

**Loopy Belief Propagation in the literature:** We would like to point out that even classical literature from the past Frey & MacKay (1997) has pointed out that while loopy belief propagation in graphs with cycles don't have known convergence guarantees, in many applications like error correction for communicating over noisy channels, belief propagation based decoding perform extremely well (abstract of Frey & MacKay (1997) makes *exactly* this case). Even considering very recent work on message passing on complex networks Newman (2023), the conclusion remains that loopy belief propagation is effective in practice but not very easily amenable to analysis. Although we showed the edges due to the similarity constraints form a cycle free graph for 1-NN based Gibbs distribution, bag constraints introduce cycles. However, Newman (2023) offers an approximate analysis based on *linearized version* of sum-product-BP that we adopt and we show that the inverse temperature parameters chosen for Adult Dataset (Table 12) are roughly orderwise within the stability region for an approximate linearized version of BP with the same graph structure as in the Gibbs distribution we define.

**Sufficient conditions for convergence of the Sum-Product Algorithm** (Mooij & Kappen, 2007):

We refer to Corollary 1 from (Mooij & Kappen, 2007) for a sufficient condition such that message updates from Loopy BP is a contraction. We substitute the values of $J_{ij}$, from our Gibbs Distribution to evaluate this condition which is: $\max_i [(|N(i)| - 1) \max_{j \in N(i)} \tanh(|J_{ij}|)] < 1$. For the case of using 1-NN constraints, $|N(i)| - 1 = B - 1$ ($B$ is the bag size). Now note that $|J_{ij}| \leq 2\lambda_b + 4\lambda_s$ as discussed above and as observed from Table 12.

And we observe that for entries in Table 12 for example for bag size 8, for Corollary 1, the condition (Eq. 15 from the paper) is satisfied and thus our LBP is a $l_\infty$-contraction and converges to a unique fixed point, irrespective of the initial messages.

For larger bags, we offer an approximate linearized BP analysis below.

**Stability analysis of the approximate linearized BP:** Now, we offer some approximate analysis on convergence of the Belief Propagation step for some simpler cases (like the Adult Dataset). In what follows, we follow the recipe given in Newman (2023) to linearize the BP. We have a Gibbs distribution over $n$ binary variables $\mathbf{y} \in \{-1, +1\}^n$ given by:

$$\mathbb{P}(\mathbf{y}) \propto \exp\left(\sum_i h_i y_i + \sum_{i \neq j} J_{ij} y_i y_j\right)$$

$h_i$ forms the external field that biases the variables away from $1/2$. However, we will analyze the Gibbs distribution without $h_i$ as this external field can be taken to be the prior bias for each of the variables.

$\{J_{ij}\}$ is very sparse and is non zero only if $i, j$ are in each others 1-NN neighborhood or $i, j$ belong to the same bag. $J_{ij}$ in our formulation could be negative but we actually, analyze the ferromagnetic version with only the correct adjacency matrix, i.e.

$$\mathbb{P}(\tilde{\mathbf{y}}) \propto \exp(-\beta \sum_{i \neq j} \mathbf{1}_{J_{ij} \neq 0} \tilde{y}_i \tilde{y}_j)$$

For the above Ising model over $\tilde{\mathbf{y}}$, we have the following normalized update rule (this is from Newman (2023)):

$$\frac{1}{2}(1 + \epsilon_{i \leftarrow j}) = \frac{1}{Z_{i \leftarrow j}} \prod_{k \in N(j) - i} \frac{1}{2} \left[ \exp(\beta)(1 + \epsilon_{j \leftarrow k}) + \exp(-\beta)(1 - \epsilon_{j \leftarrow k}) \right]$$

$$Z_{i \leftarrow j} = \sum_{r = \{+1, -1\}} \prod_{k \in N(j) - i} \frac{1}{2} \left[ \exp(\beta r)(1 + \epsilon_{j \leftarrow k}) + \exp(-\beta r)(1 - \epsilon_{j \leftarrow k}) \right] \qquad (10)$$

Here, $N(j)$ is the neighborhood of $j$ according to the graph obtained from the 1-0 adjacency matrix $1_{J_{i,j} \neq 0}$. This is equivalent to the BP iterations quoted in Section 4.1 (except for the normalization $Z_{i \leftarrow j}$ and setting $m_{i \leftarrow j}(+1) = 1 + \epsilon_{i \leftarrow j}$, $m_{i \leftarrow j}(-1) = 1 - \epsilon_{i \leftarrow j}$ due to the normalization.

Again following Newman (2023), and ignoring the second order terms in $O(\epsilon_{\{\cdot\}}^2)$, we have the following linearized Belief Propagation:

$$\epsilon_{i \leftarrow j} = \tanh \beta \sum_{k \in N(j) - i} \epsilon_{j \leftarrow k} \qquad (11)$$

We have an edge - incidence matrix $H$ of size $2|E| \times 2|E|$ where $|E|$ is the set of edges in the graph indexed by valid oriented edges $i \leftarrow j$. For the row $i \leftarrow j$, all columns corresponding to $j \leftarrow k : k \neq i,\ k \in N(j)$ have 1 and all other entries are 0.

Essentially, the messages if the linearized BP converge, then it is a fixed point of the equation: $X = \tanh(\beta) H x$. Therefore, convergence is exponentially fast if $\tanh(\beta) \|H\|_2 < 1$ as the linear operator becomes a contraction. Here, $\|H\|_2$ is the spectral norm of $H$.

**Computation of inverse temperature threshold for Adult Dataset:** For the full Adult dataset, the edge incidence matrix even for 1-NN graph is of the order of 100k. Hence, computing the largest singular value of this is a time consuming operation. However, we randomly subsampled $5k$ points from the distribution and we analyze the spectral norm of the edge incidence matrix obtained from this.

We found that $\|H\|_2 \approx 6.344$. This means that inverse temperature is at most $\tanh^{-1}(1/6.344) \approx 0.158$ for the linearized BP to converge. We observe from Table 12 that $|J_{ij}| \leq 2\lambda_b + 4\lambda_s$ is much smaller than this threshold for larger bags.

**Key Takeaway:** Exact loopy BP updates are shown to be a contraction based on sufficient conditions in Mooij & Kappen (2007) for smaller bag sizes for Adult Dataset. With an approximate linearized BP analysis of the ferromagnetic model with the same graph structure, for a subsampled set of data points i.i.d from the original Adult Dataset without replacement, we show that for large bags the chosen hyperparameters for inverse temperatures are well within the convergence threshold.

## D    LIMITATIONS AND FUTURE WORK

There are several unexplored interesting directions that we wish to pick up as future work. Notably, one of the primary ones is to explore alternate energy potentials for the Gibbs distribution other than quadratic terms we use now. It might also be of independent interest to further investigate why such a simple proposition like BP works on such a scale efficiently converging to marginals proving highly useful in supervised learning even with 1-NN based covariate information. A complete theoretical understanding behind the success of BP for the target task would be an interesting direction building on the theoretical pointers in the supplement.

