# OpenReview forum: "Learning from Label Proportions: Bootstrapping Supervised Learners via Belief Propagation"
_ICLR.cc/2024/Conference — ICLR 2024 poster_

### Official Review · Reviewer_hQ1X · 2023-10-27

**Soundness:** 3 good
**Presentation:** 3 good
**Contribution:** 2 fair
**Rating:** 5
**Confidence:** 4

**Summary:**

This paper proposes an effective and efficient approach to the problem of Learning from Label Proportions. The proposed approach has two main steps. In the first step, it uses Belief
Propagation to marginalize the Gibbs distribution to obtain pseudo labels. In the second step, it uses the pseudo labels to provide supervision for a learner. The paper conducted experiments to show the usefulness of the proposed approach.

**Strengths:**

- The problem of Learning from Label Proportions could be useful.
- This paper is well structured and easy to follow.
- The proposed approach outperforms state-of-the-art approaches.

**Weaknesses:**

- The proposed approach is complex; a simple framework is desirable.
- The proposed approach assumes the case of disjoint bags; it cannot handle non-disjoint bags.
- The paper lacks theoretical aspects of the proposed approach.

**Questions:**

The paper should discuss theoretical aspects of the proposed approach. For example, I am interested in the time and space complexity of the proposed approach since the proposed approach has rather complex framework. What are the computational and memory costs of the proposed approach?

The graph structure has a significant impact on the proposed approach. How do you determine the number of nearest neighbors? Is there any theoretical background to determine the graph structure?

As shown in Algorithm 1, the proposed approach uses the iterative computations. Does the proposed approach have a theoretical property to converge? How do you determine the number of iterations, R?

As shown in Table 1 and 2, for UCI and Criteo datasets, the proposed approach is competitive to the previous approaches. On the other hand, the proposed approach does not work well for CIFAE dataset, as shown in Table 3. Please theoretically justify the experimental results.

Since the proposed approach uses a k-NN graph, it needs a high computational time to construct the graph. In Section 6.1, the paper shows the processing time of only the proposed approach. Is the proposed approach more efficient than the previous approaches?

---

> ### Author Response · Authors · 2023-11-17
> **Response 1/2 to Reviewer hQ1X**
>
> We thank the reviewer for their detailed comments and provide clarifications below:
>
> > The proposed approach is complex; a simple framework is desirable.
>
> **Ans:** We think our algorithm is a simple 2-Step Iterative algorithm with the Belief Propagation Step and Embedding Refinement Step that are easy to implement in code. In order to be comprehensive, we detail the Belief Propagation step. Given a Ising model, there are several standard packages to perform Belief Propagation like pgMax that we use.
>
> > The proposed approach assumes the case of disjoint bags; it cannot handle non-disjoint bags.
>
> **Ans:** Our method is general enough for us to accommodate overlapping bags, the only change would be in the potential terms in the Gibbs Distribution formed, and thus the factor graph would differ from the disjoint case. This does not affect the algorithm procedure but only the outcome as expected, in fact learning non-disjoint bags should be an easier problem intuitively as more information in terms of bag constraints is available to the learner. We only discuss the disjoint bag case in the paper as it is of more practical relevance and is the one previously studied in popular literature as highlighted in Section 2. We reiterate that to run our algorithm on non-disjoint bags requires no code change and it will run out of the box when provided with non-disjoint bags as input.
>
> > The paper lacks theoretical aspects of the proposed approach.
>
> **Ans:** We would like to refer the reviewer to Section A.3 in the appendix where we provide some intuition as to why our algorithm works via showing that the nearness constraints through 1-NN produces a cycle free factor graph. However, when bags are randomly formed, invariably cycles are formed. Starting from the classical paper (Frey et al., 1997) to the very recent works on message passing (Newman, 2023), loopy belief propagation does not have convergence guarantees in general. In fact, the former points out the empirical success in decoding error correcting codes even on loopy graphs which precisely inspired us.
>
> We further provide an approximate Linearized BP analysis that shows that for large bags regime for Adult, inverse temperatures chosen actually yield convergence for the Linearized BP, in a **new Section C** in the updated manuscript. We feel this analysis of the theoretical guarantee of the algorithm’s performance helps provide some theoretical backing for our strong empirical results.
>
> > Q1: The paper should discuss theoretical aspects of the proposed approach. For example, I am interested in the time and space complexity of the proposed approach since the proposed approach has rather complex framework. What are the computational and memory costs of the proposed approach?
>
> Ans. We refer the reviewer to Section 6.1 in the main paper and section A.2.1 in the supplement for detailed discussion in the time complexity. With respect to the space complexity, we would like to mention that the only overhead for our approach is for a potential matrix for the factor graph to store the node and pairwise potentials. This only amounts to O(V+E) space complexity as we would have V unary and E pairwise terms. Note that V = data_size, and E = O((num_neighbours + bag_size) * data_size). This is because if two points are not in the K-NN neighborhood of each other and if they don’t belong together in any bag, then there would be no pairwise term corresponding to it as discussed in section 4.1. Hence the space complexity is merely O((num_neighbours + bag_size) * data_size), thus linear.
>
> > Q2: The graph structure has a significant impact on the proposed approach. How do you determine the number of nearest neighbors? Is there any theoretical background to determine the graph structure?
>
> **Ans:** The graph is an outcome of the bag constraints and the covariate pairwise constraints in the Gibbs Distribution, both of which are given to the learner. The choice of neighbors is a hyper-parameter and we have shown that even using a single Nearest Neighbor suffices in A.1.1.
>
> > Q3: As shown in Algorithm 1, the proposed approach uses the iterative computations. Does the proposed approach have a theoretical property to converge? How do you determine the number of iterations, R?
>
> **Ans:** We refer the reviewer to Section 4.3 where we define our iterative procedure and choices explicitly. To reiterate, “In principle, we could iterate it several times to refine embeddings progressively but we stop when the new iteration does not clearly improve performance on validation set“. We further refer the reviewer to section A.2.2 in the appendix where we note the performance of the algorithm on different iterations and highlight why we empirically choose iteration 2 as the stopping iteration.
>
> (1/2)

---

> ### Author Response · Authors · 2023-11-17
> **Response 2/2 to Reviewer hQ1X**
>
> > Q4: As shown in Table 1 and 2, for UCI and Criteo datasets, the proposed approach is competitive to the previous approaches. On the other hand, the proposed approach does not work well for CIFAE dataset, as shown in Table 3. Please theoretically justify the experimental results.
>
> **Ans:** As the reviewer pointed out, for the UCI and Criteo datasets our performance improvements are quite significant and outperforms all baseline methods. We would like to also reiterate that our method performs significantly well on CIFAR dataset too, as empirically demonstrated by improvements of as much as **7.4%** in AUC on larger bag sizes over the closest baseline (Table 3). An important thing to note here is that CIFAR is a much easier dataset for all algorithms as demonstrated by the closeness of the performance of all baselines to the instance wise performance which is an empirical ceiling. Due to the use of ImageNet trained SimCLR embeddings as input for all methods, the task of classification on CIFAR, especially the balanced on (CIFAR-B) is a much simpler task for all algorithms to solve and the scope for improvement is minimal. We still are competitive with, if not better than, all baselines across all bag sizes and all datasets. This is proof enough of the generalizability of our method, which is domain agnostic.
>
> > Q5: Since the proposed approach uses a k-NN graph, it needs a high computational time to construct the graph. In Section 6.1, the paper shows the processing time of only the proposed approach. Is the proposed approach more efficient than the previous approaches?
>
> **Ans:** Compared to data setup time which is common to DLLP and our method the overall algorithm time is not that significant. The training, data setup times are identical, the only overhead is the BP time. Moreover, the time taken for the kNN graph construction and the required pre-processing is merely **80.49s** (+- 6.07s) for the Adult Dataset (barely a minute for 1 nearest neighbor, for a ~50k sized dataset), which is almost 10 times smaller than the common data setup time and comparable to the training time for the MLP (**85s**). It is also in the same range as the BP time for 128 sized bags, and much faster than BP for larger bags. Thus the graph construction for kNN does not need high computational time and is not a bottleneck. As we have highlighted in section A.1.1, 1NN suffices to retrieve the majority of the performance Thus we are able to achieve a superior performance over the baselines with only additional 80s required for obtaining the neighbor graph to form our covariate factors before Belief Propagation.
>
> We hope this clears all the doubts the reviewer had. We are happy to discuss or clarify further questions as well.
>
> (2/2)

---

> ### Author Response · Authors · 2023-11-22
>
> We greatly appreciate the valuable feedback on our submission. We have done our best to answer all your questions and added a new **Section C** for theoretical guarantees for our algorithm, clarified framework and bagging, provided computational and memory costs, explained the graph structure and iterative convergence, elaborated on the computational time and experimental results to address your concerns.
>
> As the discussion deadline is nearing, we would appreciate your response to the rebuttal or any further constructive discussion.
>
> Grateful for your effort and time to review our submission!

---

### Official Review · Reviewer_XP46 · 2023-10-27

**Soundness:** 3 good
**Presentation:** 3 good
**Contribution:** 3 good
**Rating:** 6
**Confidence:** 3

**Summary:**

This paper studies the setting of learning from label proportions (LLP), where we have access to aggregate labels over bags (i.e., grouped instances). The authors provide an approach that (1) implements belief propagation to assign pseudolabels to similar data points and (2) iteratively trains supervised classifiers with the pseudolabels (and the previous classifiers learned embeddings).

**Strengths:**

* The authors provide a new scheme (based on Ising models and belief propagation) to propagate pseudolabels across datapoints taking into account bag constraints and covariate similarity.
* They also derive a new architecture and objective during bootstrapping their supervised model on the produced pseudolables. This involves an additional hidden layer that produces soft scores over the bag to maintain correct bag proportions.
* Good experimental gains over existing LLP baselines with large bag sizes.

**Weaknesses:**

1.  Lack of explanation/intuition about results. Are there any hypotheses as to why the results of your method are worse in cases with small bag sizes but better in cases with large bag sizes?


2. Lack of discussion about work from the field of weak supervision, where there have been similar problems studied in the context of combining weak supervision (labels similar to aggregate labels over bags) and covariate information via clustering [1] and via label propagation [2]. In both cases, a model is trained after pseudolabels are generated (although no iterative refinement is done as these methods start from pretrained representations and supervised models are directly fit on the pseudolabels).


3. A few typos that I noted (that don’t overall affect my score):
* Last line of page 7: “the iteration seem to help improve performance”, should be “iteration seems to help improve performance”
* “Bag constraints” in section 6.1 shouldn’t be capitalized

[1] Chen, Mayee F., et al. "Shoring up the foundations: Fusing model embeddings and weak supervision." Uncertainty in Artificial Intelligence. PMLR, 2022.
[2] Pukdee, Rattana, et al. "Label Propagation with Weak Supervision." The Eleventh International Conference on Learning Representations. 2022.

**Questions:**

* See first point in the weakness section. Are there any particular intuitions as to why DLLP outperforms your method (somewhat consistently) over small bag sizes?

---

> ### Author Response · Authors · 2023-11-17
> **Response 1/1 to Reviewer XP46**
>
> We thank the reviewer for their detailed comments and provide clarifications below:
>
> > Lack of explanation/intuition about results. Are there any hypotheses as to why the results of your method are worse in cases with small bag sizes but better in cases with large bag sizes? Are there any particular intuitions as to why DLLP outperforms your method (somewhat consistently) over small bag sizes?
>
> **Ans.** The performance of DLLP on small bags is the result of relatively higher supervision available as compared to larger bags where the number of bags can be as few as 20! (so just 20 label counts in all for the dataset) In presence of the high supervision provided by small bags, even other algorithms are able to perform comparable to our method. In case of small bag sizes our performance is not necessarily worse but in the same ballpark, within the confidence intervals, and is really close to the instance wise performance. So there is much less margin for improvement in the case of small bags as compared to large bags (*~1% for bags of size 8 vs ~10% for bags of size 2048* for Adult dataset). We would like to reiterate that even on small bags sizes we still are competitive with, if not better than, all baselines across all datasets.
>
> On large bags where the other methods collapse due to extremely weak supervision, our novel algorithm still holds up.
>
> We further provide an approximate Linearized BP analysis that shows that for large bags regime for Adult, inverse temperatures chosen actually yield convergence for the Linearized BP, in a **new Section C** in the updated manuscript. We feel this analysis of the theoretical guarantee of the algorithm’s performance helps provide the requested theoretical backing for our strong empirical results, especially on Larger Bags.
>
> > Lack of discussion about work from the field of weak supervision, where there have been similar problems studied in the context of combining weak supervision (labels similar to aggregate labels over bags) and covariate information via clustering [1] and via label propagation [2]. In both cases, a model is trained after pseudolabels are generated (although no iterative refinement is done as these methods start from pretrained representations and supervised models are directly fit on the pseudolabels).
>
> **Ans.** We have comprehensively cited all prior works in the LLP domain and compared against several strong baselines. We’ll be happy to cite and discuss other weak supervision works, but they don’t necessarily directly apply to our setup as we have a specific form of weak supervision in the form of bag level labels. Heeding the reviewer's suggestion, we have updated our manuscript’s section B in the supplement to include additional discussion on Weak Supervision and would request the reviewer to take a look at the same.
>
> > A few typos that I noted (that don’t overall affect my score):
> Last line of page 7: “the iteration seem to help improve performance”, should be “iteration seems to help improve performance”
> “Bag constraints” in section 6.1 shouldn’t be capitalized
>
> **Ans.** We thank the reviewer for pointing out these typos and have corrected them in the revised version of the manuscript. All the changes are highlighted in blue text.
>
> We hope this clears all the doubts the reviewer had. We are happy to discuss or clarify further questions as well.
>
> (1/1)

---

> > ### Comment · Reviewer_XP46 · 2023-11-19
> > **Reviewer Response**
> >
> > Thanks for your responses! I feel that most of my concerns are addressed.
> >
> > Regarding the first point: Thanks for the clarifications regarding why performance with small bags is comparable with baselines across most datasets due to the nature of the higher levels of supervision.
> >
> > Regarding the second point: I agree that the canonical (programmatic) weak supervision setup [1] does not directly apply to the LLP domain; however, many instances of weak labelers considered in that context are of the form of predicting only positive labels and otherwise abstaining (or only predicting negatives and otherwise abstaining from predictions) -- see some of the Snorkel tutorials. In that sense, many practical demonstrations of the weak supervision frameworks (e.g., Snorkel) are applied to weak labelers that provide supervision that is very similar to that of bag level labels. Regardless, I think comparisons with some of the works in this field may be interesting, although perhaps not reasonable in the time left for the rebuttal.
> >
> > [1] Ratner, Alexander J., et al. "Data programming: Creating large training sets, quickly." Advances in neural information processing systems 29 (2016).

---

> ### Author Response · Authors · 2023-11-21
> **Reply to Reviewer XP46**
>
> We thank the reviewer for the engaging discussion and are happy that our response addressed your concerns. We discuss below the weak supervision setup, specifically Snorkel in the context of LLP.
>
> Snorkel provides a way to aggregate different kinds of weak supervision sources like patterns, heuristics, external knowledge bases, crowdsourced labels, etc provided by different users (instead of crowdsourcing hard labels) and it proposes a method to synthesize labels from this information. Snorkel uses Labeling Functions that take as input a data point and assign labels (or abstain) using heuristics, pattern matching, and third-party models.
>
> In the LLP problem, aggregate labels are present for groups of instances. To use Snorkel to solve the LLP problem, we’ll need to convert these aggregate labels into Labeling Functions. Just using the aggregate labels won’t be enough since there are many ways of labeling instances so as to satisfy the aggregate label constraints. We will need to think about what other information to use to construct labeling functions. Hence, it is not clear how to directly use the snorkel approach to solve the LLP problem but it definitely looks like an interesting research direction to pursue in the future.
>
> Hope this clarifies the the doubt. We would be happy to discuss further as well. We once again thank the reviewer for their time and effort.

---

### Official Review · Reviewer_TNZ1 · 2023-10-29

**Soundness:** 2 fair
**Presentation:** 2 fair
**Contribution:** 2 fair
**Rating:** 5
**Confidence:** 3

**Summary:**

In this paper, the authors propose a method for the problem of Learning from Label Proportions (LLP), where only aggregate level labels are available for groups of instances. The proposed method incorporates both bag-level and instance-level constraints to address the LLP problem. Then, the Belief Propagation (BP) algorithm is utilized to solve the problem. Finally, the authors verify the proposed method for the  LLP Binary Classification problem.

**Strengths:**

1. The topic of this paper is interesting and important. Learning from Label Proportions (LLP) is a weak-supervised learning paradigm, which is beneficial for privacy protection.
2. The proposed method exhibits good performance on large bag sizes.

**Weaknesses:**

1. The writing of this paper needs great improvement, e.g., the connection between the LLP problem and the parity checks should be clearly elaborated. The intuition behind this should be clarified.
2. The bag-level and instance-level constraints are commonplace, making the novelty quite limited.
3. The experiments were only conducted on Binary Classification problems, while the multi-class classification is a more general case.
4. The formulation of the equations should be carefully checked, e.g., in Eq.(1), according to the definition of $y(S_i)$ (the third row in Section 3), the first term equals 0. Besides, the derivation from Eq.(1) to Eq.(2) should be provided for clear understanding.
5. The proposed method performs well on large bag sizes, but the authors do not give explanations why this is the case.
6. According to the results of Tables 4 and 5, the running time of the proposed method is far more than that of DLLP, besides, the running time of other methods is not reported.
7. According to the results of Tables 9-13, the parameters seem to need very careful fine-tuning, and the sensitivity studies of these parameters are missed.

**Questions:**

1. Why do you introduce the Gibbs distribution in the modeling? The reasons should be detailedly clarified.
2. Are the instances in each bag all labeled? If yes, then the bag level counts will equal the size of the bag.
3. Is the size of each bag in this paper equal? What if the sizes of each bag are not equal?
4. In Table 2, why not report the results of large size (512, 1024, 2048) as other tables do?
5. According to the results of Tables 1-3, it is weird that the performance of DLLP(published in 2017) is better than that of GenBags(published in 2022) and EasyLLP(published in 2023). Please explain these.

---

> ### Author Response · Authors · 2023-11-17
> **Response 1/3 to Reviewer TNZ1**
>
> We thank the reviewer for their detailed comments and provide clarifications below:
>
> > 1. The writing of this paper needs great improvement, e.g., the connection between the LLP problem and the parity checks should be clearly elaborated. The intuition behind this should be clarified.
>
> **Ans.** We would be happy to elaborate on the connection between the LLP problem and the parity check paradigm. In error correction, parity check reinforces even parity in a ‘bag’. In our case, we want to make sure it equals the bag level aggregate labels. Further, we also have pairwise constraints (parity of size 2) with nearest neighbors that impose similarity. In coding theory constraints are enforced in a hard manner. However, using the Gibbs measure (an energy function), we impose a soft version of these constraints.
>
> > 2. The bag-level and instance-level constraints are commonplace, making the novelty quite limited.
>
> **Ans.** As far as we are aware, no one has attempted to utilize the constraints in such a way (formulating a Gibbs distribution for an Ising Model, performing BP) for the LLP problem, achieving a practical and scalable solution, thus we feel this is quite a novel approach to incorporate those constraints.
>
> > 3. The experiments were only conducted on Binary Classification problems, while the multi-class classification is a more general case.
>
> **Ans.** We would like to emphasize that the binary classification LLP problem is of importance and such datasets and applications can be found in healthcare or advertising (conversion modeling). Please additionally refer to [1] below for a detailed discussion on the significance of the problem and the rising interest in the domain, especially due to privacy constraints. In fact we have demonstrated our techniques on Criteo which is a large and standard dataset in the advertising domain. Prior works like references *(Ardehaly & Culotta, 2017; Saket et al., 2022; Scott & Zhang, 2020; Busa-Fekete et al., 2023; Patrini et al., 2014)* in the manuscript do deal with only the binary LLP problem.
>
> [1] O'Brien, Conor, et al. "Challenges and approaches to privacy preserving post-click conversion prediction." arXiv preprint arXiv:2201.12666 (2022).
>
> > 4. The formulation of the equations should be carefully checked, e.g., in Eq.(1), according to the definition of $y(S_i)$ (the third row in Section 3), the first term equals 0. Besides, the derivation from Eq.(1) to Eq.(2) should be provided for clear understanding.
>
> **Ans.** The definition of $y(S_i)$ refers to the bag label, which would be the sum of the labels of all the instances comprising the bag. Note that individual instance labels are not available to the learner. The reviewer perhaps misunderstood the notation, as the first term corresponds to the least squared loss between sum of all labels in the bag and the bag label count as defined in Section 4.1 One term corresponds to the prediction and one term corresponds to the actual value, thus it denotes a typical loss function used often in ML adapted to the bag labels.
>
> The step from Eq 1 to Eq 2 is simply to replace all squared terms $(y_i)^2$ by linear teams as $y^2_i=y_i$ when $y_i$ can only take 0 or 1 values, as is the case with our energy function. We further eliminate all constant terms from the energy function as on normalization they would play no role in the distribution. We then just expand the individual terms from Eq 1 to terms reflecting the pairwise and unary potentials by simply separating out the terms with $y_i y_j$ from the terms with only $y_i$ for clarity. This helps us see the pairwise and node potentials as in Eq 3 clearly. We hope this helps elaborate the derivation of Eq 2 from Eq 1. We would be happy to add this to the updated version of the manuscript if the reviewer deems it necessary.
>
> (1/3)

---

> ### Author Response · Authors · 2023-11-17
> **Response 2/3 to Reviewer TNZ1**
>
> > 5. The proposed method performs well on large bag sizes, but the authors do not give explanations why this is the case.
>
> **Ans.** The performance of DLLP on small bags is the result of relatively higher supervision available as compared to larger bags where the number of bags can be as few as 20! (so just 20 label counts in all for the dataset) In presence of the high supervision provided by small bags, even other algorithms are able to perform comparable to our method. In case of small bag sizes our performance is not necessarily worse but in the same ballpark, within the confidence intervals, and is really close to the instance wise performance. So there is much less margin for improvement in the case of small bags as compared to large bags (**~1% for bags of size 8 vs ~10% for bags of size 2048** for Adult dataset).
>
> On large bags where the other methods collapse due to extremely weak supervision, our novel algorithm still holds up.
>
> We further provide an approximate Linearized BP analysis that shows that for large bags regime for Adult, inverse temperatures chosen actually yield convergence for the Linearized BP, in a **new Section C** in the updated manuscript. We feel this analysis of the theoretical guarantee of the algorithm’s performance helps provide the requested theoretical backing for our strong empirical results, especially on Larger Bags.
>
> > 6. According to the results of Tables 4 and 5, the running time of the proposed method is far more than that of DLLP, besides, the running time of other methods is not reported.
>
> **Ans.** We observed the time performance of other baselines, but only provided DLLP as an indicator. We have further provided the numbers for EasyLLP and GenBags on Adult and Criteo Datasets, for the training time for both algorithms, thus appending 2 columns to Table 5 Section A.2.1 and Table 4 in Section 6.1, and reinforcing the marginal overhead our method incurs. We reiterate that the Data Setup Time (Column 4) is common to all baselines and methods as well as it comprises the data loading and preprocessing times, which is identical for all methods including ours (which we have made more clear in the updated version of the manuscript). As one can see, GenBags takes more time to train the same MLP as our method (up to 5 times more on Adult and 3 times more on Criteo), and with much higher std. dev. And thus GenBags are much more time consuming, with similar or worse performance. Also, on Criteo, MLP training time is more for EasyLLP than for our method. We have updated Table 5 in the manuscript to include the 2 additional baseline’s training time as well.
>
> We reiterate that the only additional overhead as compared to the baselines is the BP step which for large bags gives enormous performance gains. We have also updated the manuscript to make this clear.
>
> > 7. According to the results of Tables 9-13, the parameters seem to need very careful fine-tuning, and the sensitivity studies of these parameters are missed.
>
> **Ans.** We did a search over the hyperparameters and ranges using Vizier (Song  et  al.,  2022) as specified in Section 5.1 and we took the best performing one through validation scores. We provide exact values in section A.6. We would like to state that the hyper-parameters are not highly sensitive as several sets of hyper parameters lead to the best performance we quote in our Tables 1, 2 and 3. For example, on Adult Dataset, Bag Size 2048 for Iteration 1 of our method along with the hyperparameter value set listed in Row 1, Table 11, as many as 44 sets of hyper-parameters lead to the AUC % reported in Table 1 for Adult on Bag size 2048. This is evidence enough to prove that the outcome of the algorithm isn’t overly dependent on the hyper-parameters, and is not very too sensitive to the choice. We only chose to report the best performing hyper parameters with a significant degree of precision to enhance exact reproducibility. We reiterate that we plan to release the code with the camera ready version of the paper.
>
> > Q.1 Why do you introduce the Gibbs distribution in the modeling? The reasons should be detailedly clarified.
>
> **Ans.** Gibbs distribution is a joint distribution over the unknown labels. By adding terms that enforce bag constraints softly and nearness constraints softly, we enforce the constraints on the label distribution. Then, we marginalize this joint distribution using BP to obtain pseudo labels which are used for the supervised learning step.
>
> (2/3)

---

> ### Author Response · Authors · 2023-11-17
> **Response 3/3 to Reviewer TNZ1**
>
> > Q.2 Are the instances in each bag all labeled? If yes, then the bag level counts will equal the size of the bag.
>
> **Ans.** No. The main setup is that we only have bag level labels, but have no access to any instance label in any bag. Hence, each bag has only one label which is equal to the mean of all instance-level labels in the bag. But we would like to train a classifier to predict instance labels on the test, thus we formulate a Gibbs Distribution and account for the uncertainty via enforcing the constraints softly. Not even one instance of the bag has labels available to the learner during training, so it’s not clear what the reviewer’s doubt is.
>
> > Q.3 Is the size of each bag in this paper equal? What if the sizes of each bag are not equal?
>
> **Ans.** Yes, while we deal with constant sized bags, the algorithm is directly applicable to the case of variable bag sizes, as only the potential terms would change in the Gibbs Distribution, and thus a different factor graph, leading to a different solution as expected. We need no algorithm change to incorporate datasets with variable bag sizes, and follow the fixed bag size setup following popular literature discussed in Section 2. We reiterate that to run our algorithm on variable sized bags requires no code change and it will run out of the box when provided with variable sized bags as input.
>
> > Q.4 In Table 2, why not report the results of large size (512, 1024, 2048) as other tables do?
>
> **Ans.** We would like to reiterate that while we tried our best to run our algorithm on Criteo for larger bag sizes, the limitation on the low-level code implementation of pgMax (Zhou et al., 2022) prohibited us from attaining any numbers. As we mention in a footnote on Page 8 of the manuscript, we were not able to run BP on Criteo for large bag sizes since we ran into integer-overflow issues. It will take some time and perhaps even involve changes to the underlying PGMax library code to resolve them to accommodate a large number of factors on Criteo which is a 1 million sized dataset!
>
> > Q.5 According to the results of Tables 1-3, it is weird that the performance of DLLP(published in 2017) is better than that of GenBags(published in 2022) and EasyLLP(published in 2023). Please explain these.
>
> **Ans.** It is true that in many cases, DLLP outperforms many results published later empirically. However, we would like to point out that GenBags, whose novelty as per the authors is “provable error bounds while being bag distribution agnostic and model agnostic” has non trivial performance in larger bag size regimes, as seen in Table 3 for CIFAR. EasyLLP is as the author's quote “a flexible and simple-to-implement debiasing approach based on aggregate labels, which operates on arbitrary loss functions” is highly competitive on smaller bag sizes on UCI Datasets, as seen in Table 1. EasyLLP and GenBags in themselves are interesting approaches to tackle this problem and have been published at prestigious conferences (NeurIPS and AISTATS respectively) and thus have been considered relevant and important by the community.
>
> We hope this clears all the doubts the reviewer had. We are happy to discuss or clarify further questions as well.
>
> (3/3)

---

> > ### Comment · Reviewer_TNZ1 · 2023-11-22
> >
> > Thanks for the response. My concerns are partially addressed. I suggest the authors make further improvements to the writing, especially the motivation and the intuition behind the method. Besides, I again suggest the authors to check the notations. (According to your definition in section 3 line 4, $y\(S_{i}\)=\sum_{j \in S_{i}} y_{j}$, then the term $\sum_{j \in S_{i}} y_{j} - y\(S_{i}\)$ in the parenthesis of the first term equals 0.) Math should be rigorous.
> >
> > I understand that binary classification has its own importance, but I would like to see the applicability of the proposed method to multi-class classification, which is a more general case. Besides, the proposed method is time-consuming compared with other methods, which is also recognized by Reviewers Svbj, c8fH. Moreover, the authors claim that the hyper-parameters are not sensitive, but they do not provide any experimental results. I  suggest the authors add the sensitivity analysis.
> >
> > For all the above reasons, I currently retain my score.

---

> ### Author Response · Authors · 2023-11-22
> **Reply to Reviewer TNZ1**
>
> We thank the reviewer for the engaging discussion and are happy that our response addressed your concerns. We provide further clarifications below.
>
> > (According to your definition in section 3 line 4, $y(S_i) = \sum_{j \in S_i} y_j$ , then the term $\sum_{j \in S_i} y_j - y(S_i)  $in the parenthesis of the first term equals 0.)
>
> **Ans.** We would like to emphasize that instance level labels are not available. Only bag level aggregates are available. Thus the usage of $y_j$ in the section 3 line 4 corresponds to true instance labels, which are not available to the learner. On the other hand, the second usage of $y_j$ in the Gibbs Distribution is for the predicted labels which are the ones on which the constraints are imposed softly via the least squared loss. We have updated the manuscript to clarify this and replaced the first usage of $y_j$ by $y^{true}_j$ in section 3, line 4.
>
> We would like to emphasize that Gibbs distributions impose constraints softly on instance wise predictions with a least squares loss between the sum of predicted labels and the true bag level sum (first term reviewer is referring to). Configurations that will have 0 loss will have low energy due to that term. Note that any configuration that matches the bag level sum will have zero loss; not necessarily the true one. Also, there could be configurations that may have non zero loss for that term but are preferred due to optimizing other constraints by our algorithm.
>
> > I understand that binary classification has its own importance, but I would like to see the applicability of the proposed method to multi-class classification, which is a more general case.
>
> **Ans.** Our primary and main contributions in this paper are for the LLP problem on binary classification problem which is also a highly relevant problem in the real world where binary labeled data is available (like in Criteo) as acknowledged by the reviewer and as evidenced by rich prior work dealing with binary classification for LLP. (Ardehaly & Culotta, 2017; Saket et al., 2022; Scott & Zhang, 2020; Busa-Fekete et al., 2023; Patrini et al., 2014). Binary Classification has applications in healthcare and advertising (conversion modelling). Please additionally refer to [1] below for a detailed discussion on the significance of the problem and the rising interest in the domain, especially due to privacy constraints.
>
> Due to reviewers request and considering the time limitation, we adapted the Gibbs measure to the multi class setting as follows. Every point has $k$ labels: $y^1_i \ldots y^k_i$ corresponding to $k$ classes.
>
> We have three main types of terms in the Gibbs measure.
>
> We impose a soft one hot constraint with the term: $(\sum_p y^p_i -1)^2$
>
> Nearness terms get modified as follows: $K(x_i,x_j) \sum_p (y^p_i - y^p_j)^2$, i.e. Euclidean distance between the vector labels of two points is small if they are nearby.
>
> Aggregate Bag level constraints have counts $b_1  \ldots b_k$. Then we simply impose a least squares constraint:
>                    $\sum_p  ( \sum_{i \in B} y^p_i - b_p)^2$
>
> All these terms are scaled by temperature hyper parameters which we search over during training. Essentially these are vectorized least squares constraints.
>
> The results on CIFAR10 in which our method is slightly better or comparable to the SOTA, are provided in Table 11 in the updated manuscript in the new section A.2.6. While we have shown the value of a quick adaptation of our work in multi class setting in the limited time, we would like to reiterate that the focus of our paper is not on the multi class setting and binary classification for an LLP is an important problem in itself as acknowledged by the reviewers as well.
>
> We hope this clarifies notation and satisfies concerns about the applicability to the multi class scenario.
>
> [1] O'Brien, Conor, et al. "Challenges and approaches to privacy preserving post-click conversion prediction." arXiv preprint arXiv:2201.12666 (2022).

---

### Official Review · Reviewer_c8fH · 2023-10-31

**Soundness:** 3 good
**Presentation:** 4 excellent
**Contribution:** 3 good
**Rating:** 8
**Confidence:** 4

**Summary:**

The paper proposes a novel algorithm for supervised learning from label proportions. One first builds a Gibbs measure that enforces the labels proportions within each bag and incentivizes nearby samples to have the same label. Then belief propagation (BP) is run on this measure, obtaining the marginals for each label. The marginals are converted to hard labels via thresholding. These new labels are then used to train a deep neural net. The network is trained on a double objective: one one side fitting the BP generated labels, on the other preserving the actual proportions of the labels in each bag. One of the internal representations of the network is then used as new covariates from which the BP and training are repeated. This algorithm achieves performances which are competitive with or superior to those of competing algorithms.

**Strengths:**

The paper is well written
The proposed algorithm is interesting and novel.
The experimental presented experimental evidence appears complete and compelling.
The algorithm achieves a good performance compares to other existing methods.

**Weaknesses:**

The proposed algorithm is slower than other algorithms it is comapred to.

**Questions:**

1. To enforce the label proportions directly in the BP have you tried sending $\lambda_b\to\infty$ and then doing MAP decoding (i.e. instead of thresholding each marginal, you take the configuration of labels that maximizes the BP approximation to the Gibbs measure)?

2. Can you provide some intuition into the architecture of the network g_L? For example what is the function of the average pooling and how it is applied.

3.This is more of a comment: BP is supposed to be more precise on sparse factor graphs. Your factor graph is not sparse due to the term with $\lambda_b$ coupling all the variables within one bag. Do you think there is a way to modify the Gibbs measure to keep the desired properties but having a sparse factor graph?

4. Can you comment on the convergence of the BP iterations? Did BP converge? did the convergence time change with the size of  the training set? Did you use some trick to make it converge?

---

> ### Author Response · Authors · 2023-11-17
> **Response 1/2 to Reviewer c8fH**
>
> We thank the reviewer for their detailed comments and provide clarifications below:
>
> > The proposed algorithm is slower than other algorithms it is compared to.
>
> **Ans:** We would like to emphasize that the running time of Belief Propagation is only linear in (bag_size + num_neighbours), and even for a dataset with 1 million samples, total run time for 128 sized bags is merely 106 minutes, where the MLP time is akin to the baselines with 10 minutes. On bag size 2048, for an adult dataset with 50k instances, it only takes 17 mins, which is negligible for the performance improvement obtained.
>
> Also, we would like to emphasize that the Data Setup time (Col 4) reported in Tables 4 & 5 includes time taken for data loading, model initialization etc. which would also contribute to DLLP total time. So DLLP total time is actually DLLP training time (Col 1) + Data Setup time for DLLP. As can be seen in the tables, DLLP training time (Col 1) is comparable to Ours - MLP training time (Col 4) as expected. So the only overhead of our method over DLLP is the BP time (Col 3) which, as clarified above, is linear in (bag_size + num_neighbours). We further refer the reviewer to Section 6.1 regarding time complexity of BP step.
>
> > Q1: To enforce the label proportions directly in the BP have you tried sending $\lambda_b \rightarrow \infty$ and then doing MAP decoding (i.e. instead of thresholding each marginal, you take the configuration of labels that maximizes the BP approximation to the Gibbs measure)?
>
> **Ans:** In a semi-supervised setting like with bag labels, we expect a lot of uncertainty on the labels per instance. From the MAP Decoding numbers (implemented by MaxProduct BP), it’s clear that attempting to label with a single label configuration is noisy and does not consistently retrieve the same performance as in the Sum Product approach across bag sizes. We have updated the manuscript by **appending analysis in Section A.2.5** about the approach suggested by the reviewer and would like to request the reviewer to take a look at the same.
>
> > Q2: Can you provide some intuition into the architecture of the network g_L? For example what is the function of the average pooling and how it is applied.
>
> **Ans:** The intuition is to obtain a single representation for the “bag”, the aggregate embedding through average polling of the instances in the bag is used to train on the bag label as in a usual supervised learning setup. As we highlight in Section 6.3, the aggregate embedding helps reinforce the bag constraint and amongst various kinds of pooling that can be done, mean pooling works the best consistently. The intuition is to simply obtain an “average” combined embedding representing the entire bag, and to obtain this one must utilize ways to combine individual embeddings of the instances comprising the bag into one single embedding. Usage of attention is one technique as we try by doing PMA as described in Section A.5. The average pooling function that our ablations from Section 6.3 point to being the best is mean pooling that is obtained by simply taking a mean of the all the instances of the bag to output one aggregate embedding for the bag via the following tensorflow code:
>
> ` pooled_embed = tf.reduce_mean(instance_embed, axis=1) `
>
> > Q3: This is more of a comment: BP is supposed to be more precise on sparse factor graphs. Your factor graph is not sparse due to the term with $\lambda_b$ coupling all the variables within one bag. Do you think there is a way to modify the Gibbs measure to keep the desired properties but having a sparse factor graph?
>
> **Ans:** It is also precise on forests, we also prove that the 1NN section of the factor graph is tree like in sec A.3 but the bags do introduce cycles. We can’t prove convergence to true marginals due to the presence of cycles in general. However, the approximation via Loopy BP is good enough as we demonstrate empirically.
>
> We would like to refer the reviewer to Section A.3 in the appendix where we provide some intuition as to why our algorithm works via showing that the nearness constraints through 1-NN produces a cycle free factor graph. However, when bags are randomly formed, invariably cycles are formed. Starting from the classical paper (Frey et al., 1997) to the very recent works on message passing (Newman, 2023), loopy belief propagation does not have convergence guarantees in general. In fact, the former points out the empirical success in decoding error correcting codes even on loopy graphs which precisely inspired us.
>
> We further provide an approximate Linearized BP analysis that shows that for large bags regime for Adult, inverse temperatures chosen actually yield convergence for the Linearized BP, in a **new Section C** in the updated manuscript.
>
> We cannot easily see a way to create a cycle free graph with bag constraints.
>
> (1/2)

---

> ### Author Response · Authors · 2023-11-17
> **Response 2/2 to Reviewer c8fH**
>
> >Q4: Can you comment on the convergence of the BP iterations? Did BP converge? did the convergence time change with the size of the training set? Did you use some trick to make it converge?
>
> **Ans:** BP usually converges in 200 iterations. We did not explicitly use any trick to converge the BP. For larger sizes of the training set, it does take more iterations to converge than for a smaller training set but we discover that 200 iterations suffice across all datasets.
>
> We hope this clears all the doubts the reviewer had. We are happy to discuss or clarify further questions as well.
>
> (2/2)

---

> > ### Comment · Reviewer_c8fH · 2023-11-22
> >
> > Thank you for your answers, which I found satisfactory. After reading them and reading other reviewers comments I confirm my score.
> >
> > I'd suggest that you include the information about BP convergence somewhere in the manuscript.

---

### Official Review · Reviewer_Svbj · 2023-11-01

**Soundness:** 2 fair
**Presentation:** 3 good
**Contribution:** 2 fair
**Rating:** 5
**Confidence:** 4

**Summary:**

This paper provided an algorithm to perform efficient learning from bag-level label proportions. The author utilized Belief Propagation on parity-like constraints derived from covariate information and bag-level constraints to obtain pseudo labels. Next, the Aggregate Embedding loss used instance-wise pseudo labels and bag-level constraints to output a final predictor. In the end, the authors also provided  experimental comparisons against several SOTA baselines across various datasets of different types.

**Strengths:**

1. Learning from bag-level Label Proportions (LLP)  is an interesting and valuable topic in the learning community. Privacy of data is one crucial consideration in this area.
2. The literature part is clear.
3. The structure of the paper is easy to follow.
4. There is extensive experiment analysis on the algorithm performance.

**Weaknesses:**

Major
1. In the setup section (section 3, p3), it lack the assumptions and descriptions on the data distribution (x,y), and bag distribution.
1.1 For example, for distributions,  the proposed algorithm may not work and or could not converge
1.2 Without data distribution assumptions, it will limit the guidance for practitioners.

2. There is no analysis of the theoretical guarantee of the algorithm's performance.

3. The proposed algorithm is much slower than the baseline algorithm, and the running time is about one order slower Table 4. However, the performance of the proposed algorithm in Table 2 and 3 are not significantly better in many setups.

4. The 4 datasets in the experimental analysis are not real data on the bag-level. The bags are manually created.

5. Some notations are not reader-friendly.
5.1 For example, in formula (2), the meaning of  | | is not defined.
5.2 3rd line in section 3 of p3, [m] is not defined.

Minor
1. In section 6.1, there is only time for one baseline algorithm, and it lacks time for other algorithms.

2. Near all parameters are tuned. There is no guidance on how to choose them in practice for quick application. For example, there is no guideline for the stopping rule of the algorithm to ensure convergence.

3. In the proposed algorithm, the pair-wise calculation could lead to a high order time complexity. For example, capturing k-nearest neighbors for every point x_i is a very slow process when the data size is large.

**Questions:**

1. What's the performance of the proposed algorithm on real bag-level data?

2. What's the time performance of other baseline algorithms?

3. Are there any stopping rules to decide when to stop the iteration and ensure the convergence?

4. What's the distribution and bagging assumption required for the proposed algorithm?

---

> ### Author Response · Authors · 2023-11-17
> **Response 1/3 to Reviewer Svbj**
>
> We thank the reviewer for their detailed comments and provide clarifications below:
>
> > In the setup section (section 3, p3), it lack the assumptions and descriptions on the data distribution (x,y), and bag distribution. 1.1 For example, for distributions, the proposed algorithm may not work and or could not converge 1.2 Without data distribution assumptions, it will limit the guidance for practitioners. *Q. 4 What's the distribution and bagging assumption required for the proposed algorithm?*
>
> **Ans.** We would like to reiterate that we do not work under any specific assumptions for the data. We only assume that the instances in each bag are sampled randomly without replacement. As mentioned in section 5, we follow the standard practice in popular LLP literature *(Chen et al, 2023; Ardehaly & Culotta, 2017; Patrini et al, 2014; Busa-Fekete et al., 2023)* of creating disjoint random bags where we sample instances without replacement from the training set, and keep repeating this for each bag, bag-size: k number of times. We are solving for the most general case of bag construction. Thus, there is no distributional assumption on the dataset or bagging.
>
> > There is no analysis of the theoretical guarantee of the algorithm's performance.
>
> **Ans** We would like to refer the reviewer to Section A.3 in the appendix where we provide some intuition as to why our algorithm works via showing that the nearness constraints through 1-NN produces a cycle free factor graph. However, when bags are randomly formed, invariably cycles are formed. Starting from the classical paper (Frey et al., 1997) to the very recent works on message passing (Newman, 2023), loopy belief propagation does not have convergence guarantees in general. In fact, the former points out the empirical success in decoding error correcting codes even on loopy graphs which precisely inspired us.
>
> We further provide an approximate Linearized BP analysis that shows that for large bags regime for Adult, inverse temperatures chosen actually yield convergence for the Linearized BP, in a **new Section C** in the updated manuscript. We feel this analysis of the theoretical guarantee of the algorithm’s performance helps provide the requested theoretical backing for our strong empirical results.
>
> > The proposed algorithm is much slower than the baseline algorithm, and the running time is about one order slower Table 4. However, the performance of the proposed algorithm in Table 2 and 3 are not significantly better in many setups.
>
> **Ans.** We would like to emphasize that the running time of Belief Propagation is only linear in (bag_size + num_neighbours), and even for a dataset with 1 million samples, total run time for 128 sized bags is merely 106 minutes, where the MLP time is akin to the baselines with 10 minutes. On bag size 2048, for an adult dataset with 50k instances, it only takes 17 mins, which is negligible for the performance improvement obtained.
>
> Also, we would like to emphasize that the Data Setup time (Col 4) reported in Tables 4 & 5 includes time taken for data loading, model initialization etc. which would also contribute to DLLP total time. So DLLP total time is actually DLLP training time (Col 1) + Data Setup time for DLLP. As can be seen in the tables, DLLP training time (Col 1) is comparable to Ours - MLP training time (Col 4) as expected. So the only overhead of our method over DLLP is the BP time (Col 3) which, as clarified above, is linear in (bag_size + num_neighbours). We further refer the reviewer to Section 6.1 regarding time complexity of BP step.
> Table 2 contains results on Criteo, which is a very hard, tabular dataset. Even small amounts of improvement on this data is non-trivial and significant. Over the last 7 years, the dataset has seen a 2% improvement in AUC while our method is able to produce up to **0.8%** improvement over DLLP *(Ardehaly & Culotta, 2017)*. In Table 3, the performance improvement in CIFAR-S is up to **7.4%**. CIFAR-B (balanced dataset) is a much easier dataset to learn from, as there are a lot of instances with label 1. Our method performs comparable to the baselines on this. The gap to the instance-wise performance is also minimal on CIFAR-B.
>
> (1/3)

---

> ### Author Response · Authors · 2023-11-17
> **Response 2/3 to Reviewer Svbj**
>
> > The 4 datasets in the experimental analysis are not real data on the bag-level. The bags are manually created. *Q.1 What's the performance of the proposed algorithm on real bag-level data?*
>
> **Ans.** We would like to refer the reviewer to several papers from the LLP literature *(Chen et al, 2023; Ardehaly & Culotta, 2017; Patrini et al, 2014; Busa-Fekete et al., 2023)* who follow the same procedure to create random bags on usual datasets. The use of Adult, Marketing, Criteo, CIFAR is well-established in the same prior literature. We request the reviewer to please take a look at Section 2 of the paper where related work from the LLP perspective is discussed. All of these prior work deal with creation of “manual” bags following some bag creation techniques.
>
> In industrial applications as well, stricter tracking regulations especially with respect to clicks as in PCM *(https://webkit.org/blog/11529/introducing-private-click-measurement-pcm/, https://developer.chrome.com/docs/privacy-sandbox/)*, have led to development of aggregation techniques with aggregation into random bags of uniform size as described by us is one of the most popular.
>
> So our setting does reflect a real-world setup and is important to study not only from a research perspective but also from an implementation perspective for future regulation ready systems.
>
> To reiterate, there is no “real bag data” that we are aware of that has been benchmarked on publicly. We would be happy to append experiments on datasets that the reviewer suggests in particular. As mentioned in section 5, we follow the standard procedure of creating disjoint random bags where we sample instances without replacement from the training set, and keep repeating this for each bag, bag-size: k number of times.
>
> > Some notations are not reader-friendly. 5.1 For example, in formula (2), the meaning of | | is not defined. 5.2 3rd line in section 3 of p3, [m] is not defined.
>
> **Ans.** We appreciate the reviewer pointing out these issues and would be happy to clarify that [m] denotes the indices {1,2,3 … m}, from which the indices corresponding to the instances in that bag S_i are obtained. | x | denotes an indicator variable which is 1 if and only if both i and j belong to the same bag. We have clarified these notations in the updated manuscript.
>
> > Near all parameters are tuned. There is no guidance on how to choose them in practice for quick application. For example, there is no guideline for the stopping rule of the algorithm to ensure convergence.
>
> **Ans.** We would like to refer the reviewer to section A.5 in the appendix where we explicitly state the default hyper-parameters and the stopping rule of the algorithm for convergence. For reproducibility purposes we also provide exact hyper-parameter values per dataset in section A.6. We would also release the code publicly with the camera ready version of the paper, further enhancing reproducibility. The choice of hyper-parameters varies from dataset to dataset, and while we cannot comment on the optimal choice of parameters for a new dataset a user might experiment on, we provide the entire implementation and experimentation detail and in near future shall provide code that we believe that should help the user extend our algorithm with ease.
>
> > In the proposed algorithm, the pair-wise calculation could lead to a high order time complexity. For example, capturing k-nearest neighbors for every point x_i is a very slow process when the data size is large.
>
> **Ans.** We refer the reviewer to Section 6.1, where we clearly state that the time complexity of Belief Propagation is linear in the number of neighbors per point. We further establish the effectiveness of 1NN in capturing the majority of the gains and thus even a small number of neighbors per point is sufficient, even when the data size is large.
>
> Further to provide empirical evidence, compared to data setup time which is common to DLLP and our method the overall algorithm time is not that significant. Moreover, the time taken for the kNN graph construction and the required pre-processing is merely 80.49s (+- 6.07s) for the Adult Dataset (barely a minute for 1 nearest neighbor, for a ~50k sized dataset), which is almost 10 times smaller than the common data setup time and comparable to the training time for the MLP (85s). It is also in the same range as the BP time for 128 sized bags, and much faster than BP for larger bags. Thus the graph construction for kNN does not need high computational time and is not a bottleneck. As we have highlighted in section A.1.1, 1NN suffices to retrieve the majority of the performance Thus we are able to achieve a superior performance over the baselines with only additional 80s required for obtaining the neighbor graph to form our covariate factors before Belief Propagation.
>
> (2/3)

---

> ### Author Response · Authors · 2023-11-17
> **Response 3/3 to Reviewer Svbj**
>
> > *Q.2 What's the time performance of other baseline algorithms?*
>
> **Ans.** We observed the time performance of other baselines, but only provided DLLP as an indicator. We have further provided the numbers for EasyLLP and GenBags on Adult and Criteo Datasets, for the training time for both algorithms, thus appending 2 columns to Table 5 Section A.2.1 and Table 4 in Section 6.1, and reinforcing the marginal overhead our method incurs. We reiterate that the Data Setup Time (Column 4) is common to all baselines and methods as well as it comprises the data loading and preprocessing times, which is identical for all methods including ours (which we have made more clear in the updated version of the manuscript). As one can see, GenBags takes more time to train the same MLP as our method (up to 5 times more on Adult and 3 times more on Criteo), and with much higher std. dev. And thus GenBags are much more time consuming, with similar or worse performance. Also, on Criteo, MLP training time is more for EasyLLP than for our method. We have updated Table 5 in the manuscript to include the 2 additional baseline’s training time as well.
>
> > *Q.3 Are there any stopping rules to decide when to stop the iteration and ensure the convergence?*
>
> **Ans.** For the BP iterations as specified in Section 5.1, we either run Belief Propagation for T = 50, 100 or 200 steps and it is observed that convergence is achieved fairly quickly and almost surely within the specified number per dataset. Even for the largest dataset (Criteo), at max 200 iterations of BP suffice for the convergence as empirically corroborated by the high AUC scores with respect to the ground truth in Section A.2.3, Table 7.
>
> For the stopping of training epochs during Step 2, which is the MLP training, we provide the exact tensorflow code snippet for early stopping in section A.5 and exact rule for halting the supervised training.
>
> For the iterations of the entire algorithm, we empirically observe that 2 iterations suffice at max to obtain consistently high performance, as evidenced in section A.2.2, Table 6. Thus we show empirically that our method converges in two iterations and no further consistent improvement is observed. We stop at the first iteration if the second iteration’s validation AUC does not outperform the first iteration’s validation AUC beyond the confidence intervals.
>
> We hope this clears all the doubts the reviewer had. We are happy to discuss or clarify further questions as well.
>
> (3/3)

---

> ### Author Response · Authors · 2023-11-22
>
> We greatly appreciate the valuable feedback on our submission. We have done our best to answer all your questions and added a new **Section C** for theoretical guarantees for our algorithm, appended additional baselines time performance to Table 4 and Table 5, clarified stopping rules, time complexity, running time, bagging scheme and clarified notations to address your concerns.
>
> As the discussion deadline is nearing, we would appreciate your response to the rebuttal or any further constructive discussion.
>
> Grateful for your effort and time to review our submission!

---

> > ### Comment · Reviewer_Svbj · 2023-11-23
> >
> > Thanks for your detailed replies. My concerns are partially addressed. Here are some of my comments:
> >
> > 1.The speed and performance issues:
> >
> > I appreciate you added two columns in Table 5. But from the table, all baseline methods have shorter running times as bag size increases. On the contrary, the proposed method becomes slower, and the gap becomes even larger (500-1000 times). This gap greatly limits the value performance improvement. In addition, in many experiments, the proposed method does not perform significantly better.
> >
> > 2. The theoretical guarantee issue
> >
> > I appreciated the added section C of theoretical analysis. Though not very rigorous, I will consider it during the next discussion stage.
> >
> > 3.The stopping rule issue,
> >
> > From Table 1 and Table 6, we can see that the performance of iterations 1, 2, and 3 vary. Sometimes, more iterations may even decrease the performance. Sometimes, even the iteration 1 is the best. Therefore, a clear stopping rule is quite important to be included.
> >
> > 4.The real data issue
> >
> > In the rebuttal, "To reiterate, there is no “real bag data” that we are aware of that has been benchmarked on publicly. " is claimed. The practitioner value is limited because no real bag data is available to test.
> >
> > Therefore, I currently retain my score. But I will consider the replies in the next discussion stage.

---

> > > ### Author Response · Authors · 2023-11-23
> > >
> > > We thank the reviewer for their follow-up discussion. We would like to clarify a few points raised in this reply.
> > >
> > > **Real Data Issue**: We would like to reiterate that for real world applications as linked in the prior reply especially with respect to clicks as in PCM (https://webkit.org/blog/11529/introducing-private-click-measurement-pcm/, https://developer.chrome.com/docs/privacy-sandbox/), aggregation techniques with aggregation into random bags of uniform size as described by us is one of the most popular. Even from a research point of view we would like to refer the reviewer to several papers from the LLP literature (Chen et al, 2023; Ardehaly & Culotta, 2017; Patrini et al, 2014; Busa-Fekete et al., 2023) who follow the same procedure to create random bags on usual datasets. The real world problem is mimicked by this random aggregation itself.
> > >
> > > **Speed and Performance Issue**: We are unsure which numbers the reviewer is referring to when they say (500-1000) times. It is important to note that the data setup time is common to all methods. So incorporating this for computing latency, the worst case scenario on Criteo for 128 sized bags in Table 4 our method in total takes 6371 seconds and DLLP, the fastest competing baseline takes 2760 seconds, barely **2.3** times more time. While for Adult dataset on 2048 sized bags in Table 5, our method takes a total of 1717 seconds, whereas EasyLLP, the fastest baseline takes 631 seconds, thus only a **2.7** times more expensive computation. This is quite marginal, especially given the fact that we only run to need the algorithm once for training and prediction is identical across all MLP based baselines. We reiterate that our **BP run time is at worst 2x compared to the data setup time** (for the largest bags for Criteo, Adult). Thus we would like to point out that the latency is not prohibitively expensive for the gain in performance.
> > >
> > > **Theoretical Guarantee**: We appreciate the reviewer appreciating the appended analysis and would like to mention that a loopy BP analysis is in general hard. General spin glass models convergence in loopy networks is challenging and beyond the scope of the paper. We do provide an approximate linearized analysis to provide insight into the theoretical guarantees our algorithm attains.
> > >
> > > **Stopping Rule**: We have clarified the stopping rule explicitly in the prior reply and re-paste it here for quick reference:
> > > For the BP iterations as specified in Section 5.1, we either run Belief Propagation for T = 50, 100 or 200 steps and it is observed that convergence is achieved fairly quickly and almost surely within the specified number per dataset. Even for the largest dataset (Criteo), at max 200 iterations of BP suffice for the convergence as empirically corroborated by the high AUC scores with respect to the ground truth in Section A.2.3, Table 7.
> > >
> > > For the stopping of epochs during Step 2, which is the MLP training, we provide the exact code for early stopping in section A.5 and the exact rule for halting the supervised training.
> > >
> > > For the iterations of the entire algorithm, we empirically observe that 2 iterations suffice at max to obtain consistently high performance, as evidenced in section A.2.2, Table 6. Thus we show empirically that our method converges in two iterations and no further consistent improvement is observed. We stop at the first iteration if the second iteration’s validation AUC does not outperform the first iteration’s validation AUC beyond the confidence intervals.
> > >
> > > We hope this clears the points raised in the replies and reinforces the effectiveness of our method. We once again thank the reviewer for their engaging discussion and time and effort in reviewing our submission.

---

> ### Author Response · Authors · 2023-11-23
>
> We thank the reviewer for their time and effort, for their reviews and discussion.
>
> **Additional Theoretical Guarantees:**
>
> We have further updated the new Section C, to add additional theoretical analysis for BP Convergence based on Sufficient conditions for convergence of the Sum-Product Algorithm (Mooij & Kappen (2007))
>
> To summarize, exact loopy BP updates are shown to be a contraction based on sufficient conditions in Mooij & Kappen (2007) for smaller bag sizes for Adult Dataset. With an approximate linearized BP analysis of the ferromagnetic model with the same graph structure, for a subsampled set of data points i.i.d from the original Adult Dataset without replacement, we show that for large bags the chosen hyperparameters for inverse temperatures are well within the convergence threshold.
>
> Thus we feel this provides the requested rigorous theoretical backing for our empirical results.

---

### Author Response · Authors · 2023-11-23
**Rebuttal Summary**

Here, we provide a succinct overview of the key updates in the manuscript and replies to the reviewers:

**Theoretical Guarantees**: Introduction of a fresh new **Section C** in the supplementary material in the updated manuscript. We provide an approximate Linearized BP analysis that shows that for large bags regime for Adult, inverse temperatures chosen actually **yield convergence** for the Linearized BP. We feel this analysis of the theoretical guarantee of the algorithm’s performance helps provide some theoretical backing for our strong empirical results although it is for an approximated version of the BP. Our analysis is based on very recent recommendations for analysis paths for loopy belief propagation.

**Running Time and Complexity** We add comparisons for the time taken for training by two additional baselines, namely EasyLLP and GenBags and update Table 4 and Table 5 to reflect the same. We reiterate that the data setup time is common to all methods, including the baselines, thus the only overhead over the baselines is the time taken for the Belief Propagation part of our algorithm which is **linear** in (bag_size + number_of_neighbors) and, as evident empirically as well, still quite manageable.

We further provide detailed clarifications for specific reviewer doubts in the individual replies. We have done our best to engage actively with reviewers and clarify all their follow-up doubts as well, as well as updated the manuscript to incorporate the reviewer feedback.

As the Author/Reviewer Discussion period deadline is coming closer it would be nice to receive reviewer feedback on our responses and engagement in discussion.

We one again thank the reviewers for their time and effort to review our submission and engage in discussion.

---

> ### Author Response · Authors · 2023-11-23
> **Additional Theoretical Analysis**
>
> We thank the reviewers for their time and effort, for their reviews and discussion.
>
> We have further updated the new **Section C**, to add **additional theoretical analysis** for BP Convergence based on Sufficient conditions for convergence of the Sum-Product Algorithm (Mooij & Kappen (2007))
>
> To summarize, exact loopy BP updates are shown to be a contraction based on sufficient conditions in Mooij & Kappen (2007) for smaller bag sizes for Adult Dataset. With an approximate linearized BP analysis of the ferromagnetic model with the same graph structure, for a subsampled set of data points i.i.d from the original Adult Dataset without replacement, we show that for large bags the chosen hyperparameters for inverse temperatures are well within the convergence threshold.
>
> Thus we feel this provides the requested rigorous theoretical backing for our empirical results.

---

### Meta-Review · Area_Chair_6Rcs · 2023-12-05

**Metareview:**

I am on the verge with this paper. While the algorithmic contribution seems nice and the problem setting mathematically interesting, I have trouble with the motivation in terms of privacy. If privacy is the key reason to reveal only label proportions then this algorithm provides a way to de-privatize the data. Isn't this a concern making the main motivation of the setting rather obsolete? I will vote for acceptance, but I would like the authors to address the concern about the motivation in the final version.

**Justification For Why Not Higher Score:**

The issue with the motivation described above makes me on verge between reject and accept, not reason to highlight.

**Justification For Why Not Lower Score:**

Some reviewers criticise lack of rigour or real data comparison, but I find the author's arguments in the rebuttal convincing on those points. I think the method is of interest.

---

### Decision · Program_Chairs · 2024-01-16

Accept (poster)